# ANOMALY TRANSFORMER: TIME SERIES ANOMALY DETECTION WITH ASSOCIATION DISCREPANCY

**Jiehui Xu**, **Haixu Wu**, **Jianmin Wang, Mingsheng Long**

School of Software, BNRist, Tsinghua University, Beijing 100084, China

`{xjh20,whx20}@mails.tsinghua.edu.cn, {jimwang,mingsheng}@tsinghua.edu.cn`

## ABSTRACT

Unsupervised detection of anomaly points in time series is a challenging problem, which requires the model to derive a distinguishable criterion. Previous methods tackle the problem mainly through learning pointwise representation or pairwise association, however, neither is sufficient to reason about the intricate dynamics. Recently, Transformers have shown great power in unified modeling of pointwise representation and pairwise association, and we find that the self-attention weight distribution of each time point can embody rich association with the whole series. Our key observation is that due to the rarity of anomalies, it is extremely difficult to build nontrivial associations from abnormal points to the whole series, thereby, the anomalies' associations shall mainly concentrate on their adjacent time points. This adjacent-concentration bias implies an association-based criterion inherently distinguishable between normal and abnormal points, which we highlight through the *Association Discrepancy*. Technically, we propose the *Anomaly Transformer* with a new *Anomaly-Attention* mechanism to compute the association discrepancy. A minimax strategy is devised to amplify the normal-abnormal distinguishability of the association discrepancy. The Anomaly Transformer achieves state-of-the-art results on six unsupervised time series anomaly detection benchmarks of three applications: service monitoring, space & earth exploration, and water treatment.

## 1 INTRODUCTION

Real-world systems always work in a continuous way, which can generate several successive measurements monitored by multi-sensors, such as industrial equipment, space probe, etc. Discovering the malfunctions from large-scale system monitoring data can be reduced to detecting the abnormal time points from time series, which is quite meaningful for ensuring security and avoiding financial loss. But anomalies are usually rare and hidden by vast normal points, making the data labeling hard and expensive. Thus, we focus on time series anomaly detection under the unsupervised setting.

Unsupervised time series anomaly detection is extremely challenging in practice. The model should learn informative representations from complex temporal dynamics through unsupervised tasks. Still, it should also derive a distinguishable criterion that can detect the rare anomalies from plenty of normal time points. Various classic anomaly detection methods have provided many unsupervised paradigms, such as the density-estimation methods proposed in local outlier factor (LOF, (Breunig et al., 2000)), clustering-based methods presented in one-class SVM (OC-SVM, (Schölkopf et al., 2001)) and SVDD (Tax & Duin, 2004). These classic methods do not consider the temporal information and are difficult to generalize to unseen real scenarios. Benefiting from the representation learning capability of neural networks, recent deep models (Su et al., 2019; Shen et al., 2020; Li et al., 2021) have achieved superior performance. A major category of methods focus on learning pointwise representations through well-designed recurrent networks and are self-supervised by the reconstruction or autoregressive task. Here, a natural and practical anomaly criterion is the pointwise reconstruction or prediction error. However, due to the rarity of anomalies, the pointwise representation is less informative for complex temporal patterns and can be dominated by normal time points, making anomalies less distinguishable. Also, the reconstruction or prediction error is calculated point by point, which cannot provide a comprehensive description of the temporal context.

---

*Equal Contribution

Another major category of methods detect anomalies based on explicit association modeling. The vector autoregression and state space models fall into this category. The graph was also used to capture the association explicitly, through representing time series with different time points as vertices and detecting anomalies by random walk (Cheng et al., 2008; 2009). In general, it is hard for these classic methods to learn informative representations and model fine-grained associations. Recently, graph neural network (GNN) has been applied to learn the dynamic graph among multiple variables in multivariate time series (Zhao et al., 2020; Deng & Hooi, 2021). While being more expressive, the learned graph is still limited to a single time point, which is insufficient for complex temporal patterns. Besides, subsequence-based methods detect anomalies by calculating the similarity among subsequences (Boniol & Palpanas, 2020). While exploring wider temporal context, these methods cannot capture the fine-grained temporal association between each time point and the whole series.

In this paper, we adapt Transfomers (Vaswani et al., 2017) to time series anomaly detection in the unsupervised regime. Transformers have achieved great progress in various areas, including natural language processing (Brown et al., 2020), machine vision (Liu et al., 2021) and time series (Zhou et al., 2021). This success is attributed to its great power in unified modeling of global representation and long-range relation. Applying Transformers to time series, we find that the temporal association of each time point can be obtained from the self-attention map, which presents as a distribution of its association weights to all the time points along the temporal dimension. The association distribution of each time point can provide a more informative description for the temporal context, indicating dynamic patterns, such as the period or trend of time series. We name the above association distribution as the *series-association*, which can be discovered from the raw series by Transformers.

Further, we observe that due to the rarity of anomalies and the dominance of normal patterns, it is harder for anomalies to build strong associations with the whole series. The associations of anomalies shall concentrate on the adjacent time points that are more likely to contain similar abnormal patterns due to the continuity. Such an adjacent-concentration inductive bias is referred to as the *prior-association*. In contrast, the dominating normal time points can discover informative associations with the whole series, not limiting to the adjacent area. Based on this observation, we try to utilize the inherent normal-abnormal distinguishability of the association distribution. This leads to a new anomaly criterion for each time point, quantified by the distance between each time point's prior-association and its series-association, named as *Association Discrepancy*. As aforementioned, because the associations of anomalies are more likely to be adjacent-concentrating, anomalies will present a smaller association discrepancy than normal time points.

Going beyond previous methods, we introduce Transformers to unsupervised time series anomaly detection and propose the *Anomaly Transformer* for association learning. To compute the Association Discrepancy, we renovate the self-attention mechanism to the *Anomaly-Attention*, which contains a two-branch structure to model the prior-association and series-association of each time point respectively. The prior-association employs the learnable Gaussian kernel to present the adjacent-concentration inductive bias of each time point, while the series-association corresponds to the self-attention weights learned from raw series. Besides, a *minimax strategy* is applied between the two branches, which can amplify the normal-abnormal distinguishability of the Association Discrepancy and further derive a new association-based criterion. Anomaly Transformer achieves strong results on six benchmarks, covering three real applications. The contributions are summarized as follows:

- Based on the key observation of Association Discrepancy, we propose the Anomaly Transformer with an Anomaly-Attention mechanism, which can model the prior-association and series-association simultaneously to embody the Association Discrepancy.

- We propose a minimax strategy to amplify the normal-abnormal distinguishability of the Association Discrepancy and further derive a new association-based detection criterion.

- Anomaly Transformer achieves the state-of-the-art anomaly detection results on six benchmarks for three real applications, justified by extensive ablations and insightful case studies.

## 2 RELATED WORK

### 2.1 UNSUPERVISED TIME SERIES ANOMALY DETECTION

As an important real-world problem, unsupervised time series anomaly detection has been widely explored. Categorizing by the anomaly determination criterion, the paradigms roughly include the density-estimation, clustering-based, reconstruction-based and autoregression-based methods.

In density-estimation methods, the classic methods such as local outlier factor (LOF, (Breunig et al., 2000)) and connectivity outlier factor (COF, (Tang et al., 2002)) respectively calculate local density and local connectivity for outlier determination. DAGMM (Zong et al., 2018) and MPPCACD (Yairi et al., 2017) integrate the Gaussian Mixture Model to estimate the density of representations.

In clustering-based methods, the anomaly score is always formalized as the distance to cluster center. SVDD (Tax & Duin, 2004) and Deep SVDD (Ruff et al., 2018) gather the representations from normal data to a compact cluster. THOC (Shen et al., 2020) fuses the multi-scale temporal features from intermediate layers by a hierarchical clustering mechanism and detects the anomalies by the multi-layer distances. ITAD (Shin et al., 2020) conducts the clustering on decomposed tensors.

The reconstruction-based models attempt to detect the anomalies by the reconstruction error. Park et al. (2018) presented the LSTM-VAE model that employs the LSTM backbone for temporal modeling and the Variational AutoEncoder (VAE) for reconstruction. OmniAnomaly proposed by Su et al. (2019) further extends the LSTM-VAE model with a normalizing flow and uses the reconstruction probabilities for detection. InterFusion from Li et al. (2021) renovates the backbone to a hierarchical VAE to model the inter- and intra-dependency among multiple series simultaneously. GANs (Goodfellow et al., 2014) are also used for reconstruction-based anomaly detection (Schlegl et al., 2019; Li et al., 2019a; Zhou et al., 2019) and perform as an adversarial regularization.

The autoregression-based models detect the anomalies by the prediction error. VAR extends ARIMA (Anderson & Kendall, 1976) and predicts the future based on the lag-dependent covariance. The autoregressive model can also be replaced by LSTMs (Hundman et al., 2018; Tariq et al., 2019).

This paper is characterized by a new association-based criterion. Different from the random walk and subsequence-based methods (Cheng et al., 2008; Boniol & Palpanas, 2020), our criterion is embodied by a co-design of the temporal models for learning more informative time-point associations.

## 2.2 TRANSFORMERS FOR TIME SERIES ANALYSIS

Recently, Transformers (Vaswani et al., 2017) have shown great power in sequential data processing, such as natural language processing (Devlin et al., 2019; Brown et al., 2020), audio processing (Huang et al., 2019) and computer vision (Dosovitskiy et al., 2021; Liu et al., 2021). For time series analysis, benefiting from the advantage of the self-attention mechanism, Transformers are used to discover the reliable long-range temporal dependencies (Kitaev et al., 2020; Li et al., 2019b; Zhou et al., 2021; Wu et al., 2021). Especially for time series anomaly detection, GTA proposed by Chen et al. (2021) employs the graph structure to learn the relationship among multiple IoT sensors, as well as the Transformer for temporal modeling and the reconstruction criterion for anomaly detection. Unlike the previous usage of Transformers, Anomaly Transformer renovates the self-attention mechanism to the Anomaly-Attention based on the key observation of association discrepancy.

## 3 METHOD

Suppose monitoring a successive system of $d$ measurements and recording the equally spaced observations over time. The observed time series $\mathcal{X}$ is denoted by a set of time points $\{x_1, x_2, \cdots, x_N\}$, where $x_t \in \mathbb{R}^d$ represents the observation of time $t$. The unsupervised time series anomaly detection problem is to determine whether $x_t$ is anomalous or not without labels.

As aforementioned, we highlight the key to unsupervised time series anomaly detection as learning informative representations and finding distinguishable criterion. We propose the *Anomaly Transformer* to discover more informative associations and tackle this problem by learning the *Association Discrepancy*, which is inherently normal-abnormal distinguishable. Technically, we propose the *Anomaly-Attention* to embody the prior-association and series-associations, along with a minimax optimization strategy to obtain a more distinguishable association discrepancy. Co-designed with the architecture, we derive an association-based criterion based on the learned association discrepancy.

## 3.1 ANOMALY TRANSFORMER

Given the limitation of Transformers (Vaswani et al., 2017) for anomaly detection, we renovate the vanilla architecture to the Anomaly Transformer (Figure 1) with an Anomaly-Attention mechanism.

**Overall Architecture** Anomaly Transformer is characterized by stacking the Anomaly-Attention blocks and feed-forward layers alternately. This stacking structure is conducive to learning underly-

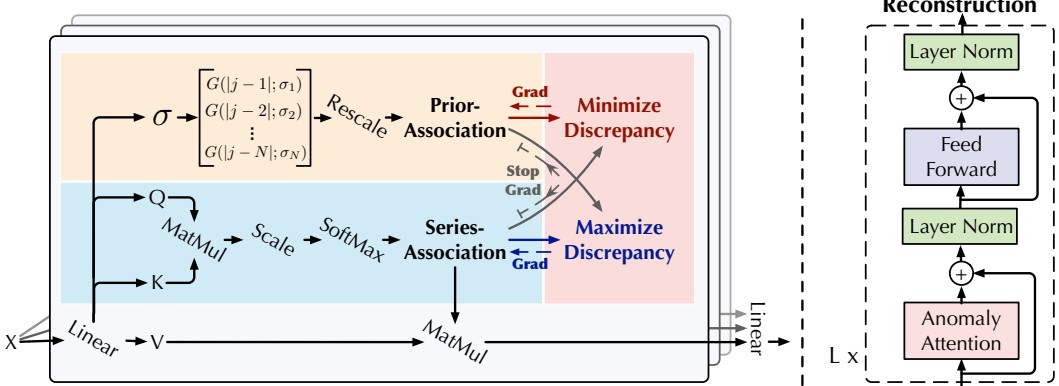

Figure 1: Anomaly Transformer. Anomaly-Attention (left) models the prior-association and series-association simultaneously. In addition to the reconstruction loss, our model is also optimized by the minimax strategy with a specially-designed stop-gradient mechanism (gray arrows) to constrain the prior- and series-associations for more distinguishable association discrepancy.

ing associations from deep multi-level features. Suppose the model contains $L$ layers with length-$N$ input time series $\mathcal{X} \in \mathbb{R}^{N \times d}$. The overall equations of the $l$-th layer are formalized as:

$$
\begin{aligned}
\mathcal{Z}^l &= \text{Layer-Norm}\Big(\text{Anomaly-Attention}(\mathcal{X}^{l-1}) + \mathcal{X}^{l-1}\Big) \\
\mathcal{X}^l &= \text{Layer-Norm}\Big(\text{Feed-Forward}(\mathcal{Z}^l) + \mathcal{Z}^l\Big),
\end{aligned}
\tag{1}
$$

where $\mathcal{X}^l \in \mathbb{R}^{N \times d_{\text{model}}}, l \in \{1, \cdots, L\}$ denotes the output of the $l$-th layer with $d_{\text{model}}$ channels. The initial input $\mathcal{X}^0 = \text{Embedding}(\mathcal{X})$ represents the embedded raw series. $\mathcal{Z}^l \in \mathbb{R}^{N \times d_{\text{model}}}$ is the $l$-th layer's hidden representation. $\text{Anomaly-Attention}(\cdot)$ is to compute the association discrepancy.

**Anomaly-Attention**  Note that the single-branch self-attention mechanism (Vaswani et al., 2017) cannot model the prior-association and series-association simultaneously. We propose the Anomaly-Attention with a two-branch structure (Figure 1). For the **prior-association**, we adopt a learnable Gaussian kernel to calculate the prior with respect to the relative temporal distance. Benefiting from the unimodal property of the Gaussian kernel, this design can pay more attention to the adjacent horizon constitutionally. We also use a learnable scale parameter $\sigma$ for the Gaussian kernel, making the prior-associations adapt to the various time series patterns, such as different lengths of anomaly segments. The **series-association** branch is to learn the associations from raw series, which can find the most effective associations adaptively. Note that these two forms maintain the temporal dependencies of each time point, which are more informative than point-wise representation. They also reflect the adjacent-concentration prior and the learned associations respectively, whose discrepancy shall be normal-abnormal distinguishable. The Anomaly-Attention in the $l$-th layer is:

$$
\begin{aligned}
\text{Initialization:} \quad & \mathcal{Q}, \mathcal{K}, \mathcal{V}, \sigma = \mathcal{X}^{l-1}W_{\mathcal{Q}}^l, \mathcal{X}^{l-1}W_{\mathcal{K}}^l, \mathcal{X}^{l-1}W_{\mathcal{V}}^l, \mathcal{X}^{l-1}W_{\sigma}^l \\
\text{Prior-Association:} \quad & \mathcal{P}^l = \text{Rescale}\left( \left[ \frac{1}{\sqrt{2\pi}\sigma_i} \exp\left( -\frac{|j-i|^2}{2\sigma_i^2} \right) \right]_{i,j \in \{1,\cdots,N\}} \right) \\
\text{Series-Association:} \quad & \mathcal{S}^l = \text{Softmax}\left( \frac{\mathcal{Q}\mathcal{K}^{\mathrm{T}}}{\sqrt{d_{\text{model}}}} \right) \\
\text{Reconstruction:} \quad & \widehat{\mathcal{Z}}^l = \mathcal{S}^l \mathcal{V},
\end{aligned}
\tag{2}
$$

where $\mathcal{Q}, \mathcal{K}, \mathcal{V} \in \mathbb{R}^{N \times d_{\text{model}}}, \sigma \in \mathbb{R}^{N \times 1}$ represent the query, key, value of self-attention and the learned scale respectively. $W_{\mathcal{Q}}^l, W_{\mathcal{K}}^l, W_{\mathcal{V}}^l \in \mathbb{R}^{d_{\text{model}} \times d_{\text{model}}}, W_{\sigma}^l \in \mathbb{R}^{d_{\text{model}} \times 1}$ represent the parameter matrices for $\mathcal{Q}, \mathcal{K}, \mathcal{V}, \sigma$ in the $l$-th layer respectively. Prior-association $\mathcal{P}^l \in \mathbb{R}^{N \times N}$ is generated based on the learned scale $\sigma \in \mathbb{R}^{N \times 1}$ and the $i$-th element $\sigma_i$ corresponds to the $i$-th time point. Concretely, for the $i$-th time point, its association weight to the $j$-th point is calculated by the Gaussian kernel $G(|j-i|; \sigma_i) = \frac{1}{\sqrt{2\pi}\sigma_i} \exp(-\frac{|j-i|^2}{2\sigma_i^2})$ w.r.t. the distance $|j-i|$. Further, we use $\text{Rescale}(\cdot)$ to transform the association weights to discrete distributions $\mathcal{P}^l$ by dividing the row sum. $\mathcal{S}^l \in \mathbb{R}^{N \times N}$

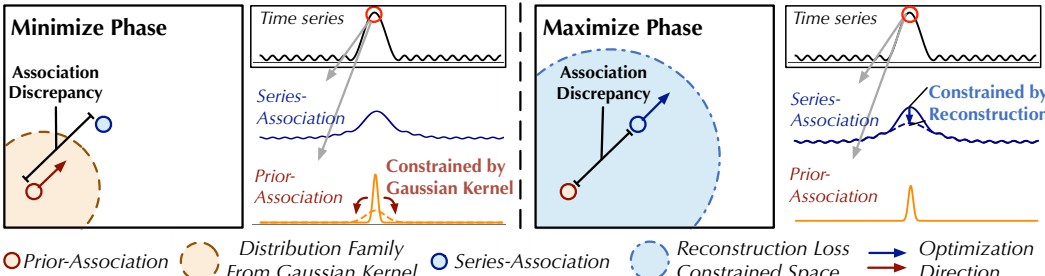

Figure 2: Minimax association learning. At the minimize phase, the prior-association minimizes the Association Discrepancy within the distribution family derived by Gaussian kernel. At the maximize phase, the series-association maximizes the Association Discrepancy under the reconstruction loss.

denotes the series-associations. $\mathrm{Softmax}(\cdot)$ normalizes the attention map along the last dimension, and each row of $\mathcal{S}^l$ forms a discrete distribution. $\widehat{\mathcal{Z}}^l \in \mathbb{R}^{N \times d_{\text{model}}}$ is the hidden representation after the Anomaly-Attention in the $l$-th layer. We use $\mathrm{Anomaly\text{-}Attention}(\cdot)$ to summarize Equation 2. In the multi-head version, the learned scale is $\sigma \in \mathbb{R}^{N \times h}$ for $h$ heads. $\mathcal{Q}_m, \mathcal{K}_m, \mathcal{V}_m \in \mathbb{R}^{N \times \frac{d_{\text{model}}}{h}}$ denote the query, key and value of the $m$-th head respectively. The block concatenates the outputs $\{\widehat{\mathcal{Z}}_m^l \in \mathbb{R}^{N \times \frac{d_{\text{model}}}{h}}\}_{1 \le m \le h}$ from multiple heads and gets the final result $\widehat{\mathcal{Z}}^l \in \mathbb{R}^{N \times d_{\text{model}}}$.

**Association Discrepancy**  We formalize the *Association Discrepancy* as the symmetrized KL divergence between prior- and series-associations, which represents the information gain between these two distributions (Neal, 2007). We average the association discrepancy from multiple layers to combine the associations from multi-level features into a more informative measure as:

$$\mathrm{AssDis}(\mathcal{P}, \mathcal{S}; \mathcal{X}) = \left[ \frac{1}{L} \sum_{l=1}^{L} \left( \mathrm{KL}(\mathcal{P}_{i,:}^l \| \mathcal{S}_{i,:}^l) + \mathrm{KL}(\mathcal{S}_{i,:}^l \| \mathcal{P}_{i,:}^l) \right) \right]_{i=1,\cdots,N} \tag{3}$$

where $\mathrm{KL}(\cdot \| \cdot)$ is the KL divergence computed between two discrete distributions corresponding to every row of $\mathcal{P}^l$ and $\mathcal{S}^l$. $\mathrm{AssDis}(\mathcal{P}, \mathcal{S}; \mathcal{X}) \in \mathbb{R}^{N \times 1}$ is the point-wise association discrepancy of $\mathcal{X}$ with respect to prior-association $\mathcal{P}$ and series-association $\mathcal{S}$ from multiple layers. The $i$-th element of AssDis corresponds to the $i$-th time point of $\mathcal{X}$. From previous observation, anomalies will present smaller $\mathrm{AssDis}(\mathcal{P}, \mathcal{S}; \mathcal{X})$ than normal time points, which makes AssDis inherently distinguishable.

## 3.2 Minimax Association Learning

As an unsupervised task, we employ the reconstruction loss for optimizing our model. The reconstruction loss will guide the series-association to find the most informative associations. To further amplify the difference between normal and abnormal time points, we also use an additional loss to enlarge the association discrepancy. Due to the unimodal property of the prior-association, the discrepancy loss will guide the series-association to pay more attention to the non-adjacent area, which makes the reconstruction of anomalies harder and makes anomalies more identifiable. The loss function for input series $\mathcal{X} \in \mathbb{R}^{N \times d}$ is formalized as:

$$\mathcal{L}_{\text{Total}}(\widehat{\mathcal{X}}, \mathcal{P}, \mathcal{S}, \lambda; \mathcal{X}) = \|\mathcal{X} - \widehat{\mathcal{X}}\|_{\text{F}}^2 - \lambda \times \|\mathrm{AssDis}(\mathcal{P}, \mathcal{S}; \mathcal{X})\|_1 \tag{4}$$

where $\widehat{\mathcal{X}} \in \mathbb{R}^{N \times d}$ denotes the reconstruction of $\mathcal{X}$. $\|\cdot\|_{\text{F}}, \|\cdot\|_k$ indicate the Frobenius and $k$-norm. $\lambda$ is to trade off the loss terms. When $\lambda > 0$, the optimization is to enlarge the association discrepancy. A minimax strategy is proposed to make the association discrepancy more distinguishable.

**Minimax Strategy**  Note that directly maximizing the association discrepancy will extremely reduce the scale parameter of the Gaussian kernel (Neal, 2007), making the prior-association meaningless. Towards a better control of association learning, we propose a minimax strategy (Figure 2). Concretely, for the **minimize phase**, we drive the prior-association $\mathcal{P}^l$ to approximate the series-association $\mathcal{S}^l$ that is learned from raw series. This process will make the prior-association adapt to various temporal patterns. For the **maximize phase**, we optimize the series-association to enlarge the association discrepancy. This process forces the series-association to pay more attention to the

non-adjacent horizon. Thus, integrating the reconstruction loss, the loss functions of two phases are:

$$\text{Minimize Phase: } \mathcal{L}_{\text{Total}}(\widehat{\mathcal{X}}, \mathcal{P}, \mathcal{S}_{\text{detach}}, -\lambda; \mathcal{X})$$
$$\text{Maximize Phase: } \mathcal{L}_{\text{Total}}(\widehat{\mathcal{X}}, \mathcal{P}_{\text{detach}}, \mathcal{S}, \lambda; \mathcal{X}),$$
(5)

where $\lambda > 0$ and $*_{\text{detach}}$ means to stop the gradient backpropagation of the association (Figure 1). As $\mathcal{P}$ approximates $\mathcal{S}_{\text{detach}}$ in the minimize phase, the maximize phase will conduct a stronger constraint to the series-association, forcing the time points to pay more attention to the non-adjacent area. Under the reconstruction loss, this is much harder for anomalies to achieve than normal time points, thereby amplifying the normal-abnormal distinguishability of the association discrepancy.

**Association-based Anomaly Criterion**   We incorporate the normalized association discrepancy to the reconstruction criterion, which will take the benefits of both temporal representation and the distinguishable association discrepancy. The final anomaly score of $\mathcal{X} \in \mathbb{R}^{N \times d}$ is shown as follows:

$$\text{AnomalyScore}(\mathcal{X}) = \text{Softmax}\Big( - \text{AssDis}(\mathcal{P}, \mathcal{S}; \mathcal{X}) \Big) \odot \Big[ \|\mathcal{X}_{i,:} - \widehat{\mathcal{X}}_{i,:}\|_2^2 \Big]_{i=1,\cdots,N}$$
(6)

where $\odot$ is the element-wise multiplication. $\text{AnomalyScore}(\mathcal{X}) \in \mathbb{R}^{N \times 1}$ denotes the point-wise anomaly criterion of $\mathcal{X}$. Towards a better reconstruction, anomalies usually decrease the association discrepancy, which will still derive a higher anomaly score. Thus, this design can make the reconstruction error and the association discrepancy collaborate to improve detection performance.

## 4 EXPERIMENTS

We extensively evaluate Anomaly Transformer on six benchmarks for three practical applications.

**Datasets**   Here is a description of the six experiment datasets: (1) SMD (Server Machine Dataset, Su et al. (2019)) is a 5-week-long dataset collected from a large Internet company with 38 dimensions. (2) PSM (Pooled Server Metrics, Abdulaal et al. (2021)) is collected internally from multiple application server nodes at eBay with 26 dimensions. (3) Both MSL (Mars Science Laboratory rover) and SMAP (Soil Moisture Active Passive satellite) are public datasets from NASA (Hundman et al., 2018) with 55 and 25 dimensions respectively, which contain the telemetry anomaly data derived from the Incident Surprise Anomaly (ISA) reports of spacecraft monitoring systems. (4) SWaT (Secure Water Treatment, Mathur & Tippenhauer (2016)) is obtained from 51 sensors of the critical infrastructure system under continuous operations. (5) NeurIPS-TS (NeurIPS 2021 Time Series Benchmark) is a dataset proposed by Lai et al. (2021) and includes five time series anomaly scenarios categorized by behavior-driven taxonomy as point-global, pattern-contextual, pattern-shapelet, pattern-seasonal and pattern-trend. The statistical details are summarized in Table 13 of Appendix.

**Implementation details**   Following the well-established protocol in Shen et al. (2020), we adopt a non-overlapped sliding window to obtain a set of sub-series. The sliding window is with a fixed size of 100 for all datasets. We label the time points as anomalies if their anomaly scores (Equation 6) are larger than a certain threshold $\delta$. The threshold $\delta$ is determined to make a proportion $r$ of time points of the validation dataset labeled as anomalies. For the main results, we set $r = 0.1\%$ for SWaT, 0.5% for SMD and 1% for other datasets. We adopt the widely-used adjustment strategy (Xu et al., 2018; Su et al., 2019; Shen et al., 2020): if a time point in a certain successive abnormal segment is detected, all anomalies in this abnormal segment are viewed to be correctly detected. This strategy is justified from the observation that an abnormal time point will cause an alert and further make the whole segment noticed in real-world applications. Anomaly Transformer contains 3 layers. We set the channel number of hidden states $d_{\text{model}}$ as 512 and the number of heads $h$ as 8. The hyperparameter $\lambda$ (Equation 4) is set as 3 for all datasets to trade-off two parts of the loss function. We use the ADAM (Kingma & Ba, 2015) optimizer with an initial learning rate of $10^{-4}$. The training process is early stopped within 10 epochs with the batch size of 32. All the experiments are implemented in Pytorch (Paszke et al., 2019) with a single NVIDIA TITAN RTX 24GB GPU.

**Baselines**   We extensively compare our model with 18 baselines, including the reconstruction-based models: InterFusion (2021), BeatGAN (2019), OmniAnomaly (2019), LSTM-VAE (2018); the density-estimation models: DAGMM (2018), MPPCACD (2017), LOF (2000); the clustering-based methods: ITAD (2020), THOC (2020), Deep-SVDD (2018); the autoregression-based models: CL-MPPCA (2019), LSTM (2018), VAR (1976); the classic methods: OC-SVM (2004), IsolationForest (2008). Another 3 baselines from change point detection and time series segmentation are deferred to Appendix I. InterFusion (2021) and THOC (2020) are the state-of-the-art deep models.

Figure 3: ROC curves (horizontal-axis: false-positive rate; vertical-axis: true-positive rate) for the five datasets. A higher AUC value (area under the ROC curve) indicates a better performance. The predefined threshold proportion $r$ is in $\{0.5\%, 1.0\%, 1.5\%, 2.0\%, 10\%, 20\%, 30\%\}$.

Table 1: Quantitative results for Anomaly Transformer (*Ours*) in the five datasets. The *P*, *R* and *F1* represent the precision, recall and F1-score (as %) respectively. F1-score is the harmonic mean of precision and recall. For these three metrics, a higher value indicates a better performance.

| Dataset | SMD | | | MSL | | | SMAP | | | SWaT | | | PSM | | |
|---|---|---|---|---|---|---|---|---|---|---|---|---|---|---|---|
| Metric | P | R | F1 | P | R | F1 | P | R | F1 | P | R | F1 | P | R | F1 |
| OCSVM | 44.34 | 76.72 | 56.19 | 59.78 | 86.87 | 70.82 | 53.85 | 59.07 | 56.34 | 45.39 | 49.22 | 47.23 | 62.75 | 80.89 | 70.67 |
| IsolationForest | 42.31 | 73.29 | 53.64 | 53.94 | 86.54 | 66.45 | 52.39 | 59.07 | 55.53 | 49.29 | 44.95 | 47.02 | 76.09 | 92.45 | 83.48 |
| LOF | 56.34 | 39.86 | 46.68 | 47.72 | 85.25 | 61.18 | 58.93 | 56.33 | 57.60 | 72.15 | 65.43 | 68.62 | 57.89 | 90.49 | 70.61 |
| Deep-SVDD | 78.54 | 79.67 | 79.10 | 91.92 | 76.63 | 83.58 | 89.93 | 56.02 | 69.04 | 80.42 | 84.45 | 82.39 | 95.41 | 86.49 | 90.73 |
| DAGMM | 67.30 | 49.89 | 57.30 | 89.60 | 63.93 | 74.62 | 86.45 | 56.73 | 68.51 | 89.92 | 57.84 | 70.40 | 93.49 | 70.03 | 80.08 |
| MMPCACD | 71.20 | 79.28 | 75.02 | 81.42 | 61.31 | 69.95 | 88.61 | 75.84 | 81.73 | 82.52 | 68.29 | 74.73 | 76.26 | 78.35 | 77.29 |
| VAR | 78.35 | 70.26 | 74.08 | 74.68 | 81.42 | 77.90 | 81.38 | 53.88 | 64.83 | 81.59 | 60.29 | 69.34 | 90.71 | 83.82 | 87.13 |
| LSTM | 78.55 | 85.28 | 81.78 | 85.45 | 82.50 | 83.95 | 89.41 | 78.13 | 83.39 | 86.15 | 83.27 | 84.69 | 76.93 | 89.64 | 82.80 |
| CL-MPPCA | 82.36 | 76.07 | 79.09 | 73.71 | 88.54 | 80.44 | 86.13 | 63.16 | 72.88 | 76.78 | 81.50 | 79.07 | 56.02 | 99.93 | 71.80 |
| ITAD | 86.22 | 73.71 | 79.48 | 69.44 | 84.09 | 76.07 | 82.42 | 66.89 | 73.85 | 63.13 | 52.08 | 57.08 | 72.80 | 64.02 | 68.13 |
| LSTM-VAE | 75.76 | 90.08 | 82.30 | 85.49 | 79.94 | 82.62 | 92.20 | 67.75 | 78.10 | 76.00 | 89.50 | 82.20 | 73.62 | 89.92 | 80.96 |
| BeatGAN | 72.90 | 84.09 | 78.10 | 89.75 | 85.42 | 87.53 | 92.38 | 55.85 | 69.61 | 64.01 | 87.46 | 73.92 | 90.30 | 93.84 | 92.04 |
| OmniAnomaly | 83.68 | 86.82 | 85.22 | 89.02 | 86.37 | 87.67 | 92.49 | 81.99 | 86.92 | 81.42 | 84.30 | 82.83 | 88.39 | 74.46 | 80.83 |
| InterFusion | 87.02 | 85.43 | 86.22 | 81.28 | 92.70 | 86.62 | 89.77 | 88.52 | 89.14 | 80.59 | 85.58 | 83.01 | 83.61 | 83.45 | 83.52 |
| THOC | 79.76 | 90.95 | 84.99 | 88.45 | 90.97 | 89.69 | 92.06 | 89.34 | 90.68 | 83.94 | 86.36 | 85.13 | 88.14 | 90.99 | 89.54 |
| Ours | 89.40 | 95.45 | **92.33** | 92.09 | 95.15 | **93.59** | 94.13 | 99.40 | **96.69** | 91.55 | 96.73 | **94.07** | 96.91 | 98.90 | **97.89** |

## 4.1 MAIN RESULTS

**Real-world datasets** We extensively evaluate our model on five real-world datasets with ten competitive baselines. As shown in Table 1, Anomaly Transformer achieves the consistent state-of-the-art on all benchmarks. We observe that deep models that consider the temporal information outperform the general anomaly detection model, such as Deep-SVDD (Ruff et al., 2018) and DAGMM (Zong et al., 2018), which verifies the effectiveness of temporal modeling. Our proposed Anomaly Transformer goes beyond the point-wise representation learned by RNNs and models the more informative associations. The results in Table 1 are persuasive for the advantage of association learning in time series anomaly detection. In addition, we plot the ROC curve in Figure 3 for a complete comparison. Anomaly Transformer has the highest AUC values on all five datasets. It means that our model performs well in the false-positive and true-positive rates under various preset thresholds, which is important for real-world applications.

**NeurIPS-TS benchmark** This benchmark is generated from well-designed rules proposed by Lai et al. (2021), including all types of anomalies and covering both the point-wise and pattern-wise anomalies. As shown in Figure 4, Anomaly Transformer can still achieve state-of-the-art performance. This verifies the effectiveness of our model on various anomalies.

**Ablation study** As shown in Table 2, we further investigate the effect of each part in our model. Our association-based criterion outperforms the widely-used reconstruction criterion consistently.

Figure 4: Results for NeurIPS-TS.

Specifically, the association-based criterion brings a remarkable 18.76% (76.20→94.96) averaged absolute F1-score promotion. Also, directly taking the association discrepancy as the criterion still achieves a good performance (F1-score: 91.55%) and surpasses the previous state-of-the-art model

THOC (F1-score: 88.01% calculated from Table 1). Besides, the learnable prior-association (corresponding to $\sigma$ in Equation 2) and the minimax strategy can further improve our model and get 8.43% (79.05→87.48) and 7.48% (87.48→94.96) averaged absolute promotions respectively. Finally, our proposed Anomaly Transformer surpasses the pure Transformer by 18.34% (76.62→94.96) absolute improvement. These verify that each module of our design is effective and necessary. More ablations of association discrepancy can be found in Appendix D.

Table 2: Ablation results (F1-score) in anomaly criterion, prior-association and optimization strategy. *Recon*, *AssDis* and *Assoc* mean the pure reconstruction performance, pure association discrepancy and our proposed association-based criterion respectively. *Fix* is to fix *Learnable* scale parameter $\sigma$ of prior-association as 1.0. *Max* and *Minimax* refer to the strategies for association discrepancy in the maximization (Equation 4) and minimax (Equation 5) way respectively.

| Architecture | Anomaly Criterion | Prior-Association | Optimization Strategy | SMD | MSL | SMAP | SWaT | PSM | Avg F1 (as %) |
|---|---|---|---|---|---|---|---|---|---|
| Transformer | Recon | × | × | 79.72 | 76.64 | 73.74 | 74.56 | 78.43 | 76.62 |
| Anomaly Transformer | Recon | Learnable | Minmax | 71.35 | 78.61 | 69.12 | 81.53 | 80.40 | 76.20 |
| | AssDis | Learnable | Minmax | 87.57 | 90.50 | 90.98 | 93.21 | 95.47 | 91.55 |
| | Assoc | Fix | Max | 83.95 | 82.17 | 70.65 | 79.46 | 79.04 | 79.05 |
| | Assoc | Learnable | Max | 88.88 | 85.20 | 87.84 | 81.65 | 93.83 | 87.48 |
| *final | Assoc | Learnable | Minmax | **92.33** | **93.59** | **96.90** | **94.07** | **97.89** | **94.96** |

## 4.2 Model Analysis

To explain how our model works intuitively, we provide the visualization and statistical results for our three key designs: anomaly criterion, learnable prior-association and optimization strategy.

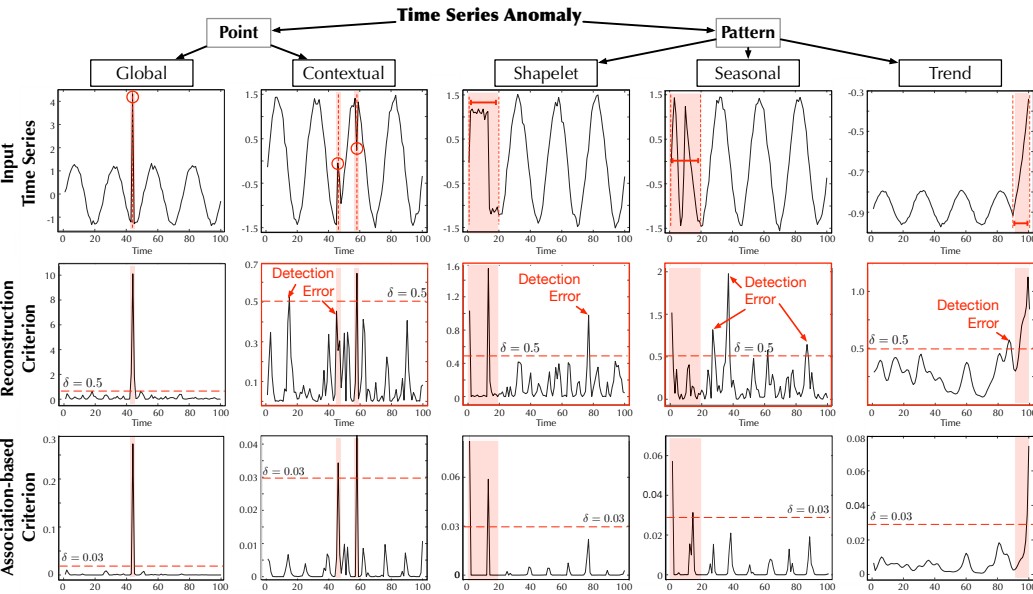

Figure 5: Visualization of different anomaly categories (Lai et al., 2021). We plot the raw series (first row) from NeurIPS-TS dataset, as well as their corresponding reconstruction (second row) and association-based criteria (third row). The point-wise anomalies are marked by red circles and the pattern-wise anomalies are in red segments. The wrongly detected cases are bounded by red boxes.

**Anomaly criterion visualization** To get more intuitive cases about how association-based criterion works, we provide some visualization in Figure 5 and explore the criterion performance under different types of anomalies, where the taxonomy is from Lai et al. (2021). We can find that our proposed association-based criterion is more distinguishable in general. Concretely, the association-based criterion can obtain the consistent smaller values for the normal part, which is quite contrasting

in point-contextual and pattern-seasonal cases (Figure 5). In contrast, the jitter curves of the reconstruction criterion make the detection process confused and fail in the aforementioned two cases. This verifies that our criterion can highlight the anomalies and provide distinct values for normal and abnormal points, making the detection precise and reducing the false-positive rate.

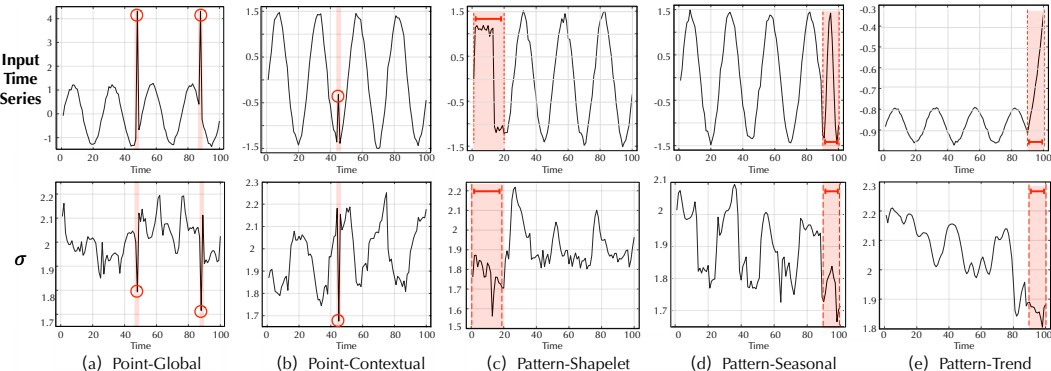

Figure 6: Learned scale parameter $\sigma$ for different types of anomalies (highlight in red).

**Prior-association visualization**  During the minimax optimization, the prior-association is learned to get close to the series-association. Thus, the learned $\sigma$ can reflect the adjacent-concentrating degree of time series. As shown in Figure 6, we find that $\sigma$ changes to adapt to various data patterns of time series. Especially, the prior-association of anomalies generally has a smaller $\sigma$ than normal time points, which matches our adjacent-concentration inductive bias of anomalies.

**Optimization strategy analysis**  Only with the reconstruction loss, the abnormal and normal time points present similar behavior in the association weights to adjacent time points, corresponding to a contrast value closed to 1 (Table 3). Maximizing the association discrepancy will force the series-associations to pay more attention to the non-adjacent area. However, to obtain a better reconstruction, the anomalies must maintain much larger adjacent association weights than normal time points, corresponding to a larger contrast value. But direct maximization will cause optimization difficulty of Gaussian kernel, and cannot strongly amplify the difference between normal and abnormal time points as expected (SMD:1.15→1.27). The minimax strategy optimizes the prior-association to provide a stronger constraint to series-association, thereby obtaining more distinguishable contrast values and better performance than the direct maximization (SMD:1.27→2.39).

Table 3: Results of adjacent association weights for *Abnormal* and *Normal* time points respectively. *Recon*, *Max* and *Minimax* represent the association learning process that is supervised by reconstruction loss, direct maximization and minimax strategy respectively. A higher contrast value ($\frac{\text{Abnormal}}{\text{Normal}}$) indicates a stronger distinguishability between normal and abnormal time points.

| Dataset | SMD | | | MSL | | | SMAP | | | SWaT | | | PSM | | |
|---|---|---|---|---|---|---|---|---|---|---|---|---|---|---|---|
| Optimization | Recon | Max | Ours | Recon | Max | Ours | Recon | Max | Ours | Recon | Max | Ours | Recon | Max | Ours |
| Abnormal (%) | 1.08 | 0.95 | 0.86 | 1.01 | 0.65 | 0.35 | 1.29 | 1.18 | 0.70 | 1.27 | 0.89 | 0.37 | 1.02 | 0.56 | 0.29 |
| Normal (%) | 0.94 | 0.75 | 0.36 | 1.00 | 0.59 | 0.22 | 1.23 | 1.09 | 0.49 | 1.18 | 0.78 | 0.21 | 0.99 | 0.54 | 0.11 |
| Contrast ($\frac{\text{Abnormal}}{\text{Normal}}$) | 1.15 | 1.27 | **2.39** | 1.01 | 1.10 | **1.59** | 1.05 | 1.08 | **1.43** | 1.08 | 1.14 | **1.76** | 1.03 | 1.04 | **2.64** |

## 5  CONCLUSION AND FUTURE WORK

This paper studies the unsupervised time series anomaly detection problem. Unlike previous works, we learn the more informative time-point associations by Transformers. Based on the key observation of association discrepancy, we propose the Anomaly Transformer, including an Anomaly-Attention with the two-branch structure to embody the association discrepancy. A minimax strategy is adopted to further amplify the difference between normal and abnormal time points. By introducing the association discrepancy, we propose the association-based criterion, which makes the reconstruction performance and association discrepancy collaborate. Anomaly Transformer achieves the state-of-the-art results on an exhaustive set of empirical studies. Future work includes theoretical study of Anomaly Transformer in light of classic analysis for autoregression and state space models.

ACKNOWLEDGMENTS

This work was supported by the National Megaproject for New Generation AI (2020AAA0109201), National Natural Science Foundation of China (62022050 and 62021002), Beijing Nova Program (Z201100006820041), and BNRist Innovation Fund (BNR2021RC01002).

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

## A    PARAMETER SENSITIVITY

We set the window size as 100 throughout the main text, which considers the temporal information, memory and computation efficiency. And we set the loss weight $\lambda$ based on the convergence property of the training curve.

Furthermore, Figure 7 provides the model performance under different choices of the window size and the loss weight. We present that our model is stable to the window size over extensive datasets (Figure 7 left). Note that a larger window size indicates a larger memory cost and a smaller sliding number. Especially, only considering the performance, its relationship to the window size can be determined by the data pattern. For example, our model performs better when the window size is 50 for the SMD dataset. Besides, we adopt the loss weight $\lambda$ in Equation 5 to trade off the reconstruction loss and the association part. We find that $\lambda$ is stable and easy to tune in the range of 2 to 4. The above results verify the sensitivity of our model, which is essential for applications.

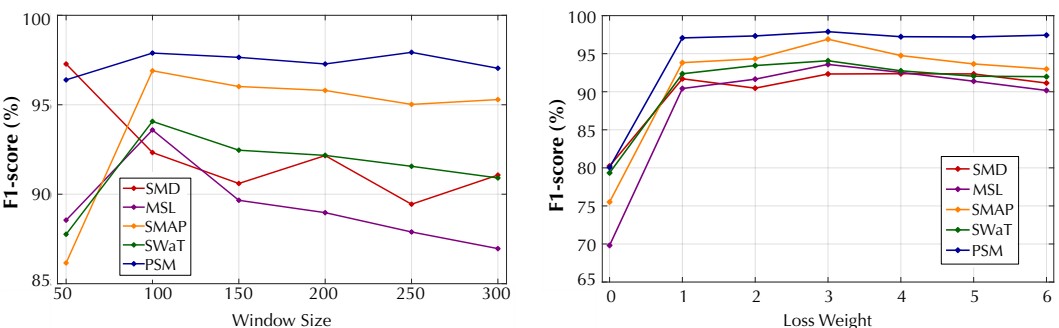

Figure 7: Parameter sensitivity for sliding window size (left) and loss weight $\lambda$ (right). The model with $\lambda = 0$ still adopts the association-based criterion but only supervised by reconstruction loss.

## B    IMPLEMENTATION DETAILS

We present the pseudo-code of Anomaly-Attention in Algorithm 1.

---
**Algorithm 1** Anomaly-Attention Mechanism (multi-head version).

---

**Input:** $\mathcal{X} \in \mathbb{R}^{N \times d_{\text{model}}}$: input; $\mathcal{D} = \left( (j-i)^2 \right)_{i,j \in \{1,\cdots,N\}} \in \mathbb{R}^{N \times N}$: relative distance matrix

**Layer params:** $\texttt{MLP}_{\texttt{input}}$: linear projector for input; $\texttt{MLP}_{\texttt{output}}$: linear projector for output

1: $\mathcal{Q},\mathcal{K},\mathcal{V},\sigma = \texttt{Split}\left(\texttt{MLP}_{\texttt{input}}(\mathcal{X}), \texttt{dim=1}\right)$ $\qquad\qquad\triangleright \mathcal{Q},\mathcal{K},\mathcal{V} \in \mathbb{R}^{N \times d_{\text{model}}}, \sigma \in \mathbb{R}^{N \times h}$

2: **for** $(\mathcal{Q}_m,\mathcal{K}_m,\mathcal{V}_m,\sigma_m)$ **in** $(\mathcal{Q},\mathcal{K},\mathcal{V},\sigma)$**:** $\qquad\triangleright \mathcal{Q}_m,\mathcal{K}_m,\mathcal{V}_m \in \mathbb{R}^{N \times \frac{d_{\text{model}}}{h}}, \sigma_m \in \mathbb{R}^{N \times 1}$

3: $\qquad \sigma_m = \texttt{Broadcast}(\sigma_m, \texttt{dim=1})$ $\qquad\qquad\qquad\qquad\qquad\triangleright \sigma_m \in \mathbb{R}^{N \times N}$

4: $\qquad \mathcal{P}_m = \frac{1}{\sqrt{2\pi}\sigma_m} \exp\left(-\frac{\mathcal{D}}{2\sigma_m^2}\right)$ $\qquad\qquad\qquad\qquad\triangleright \mathcal{P}_m \in \mathbb{R}^{N \times N}$

5: $\qquad \mathcal{P}_m = \mathcal{P}_m/\texttt{Broadcast}\left(\texttt{Sum}(\mathcal{P}_m, \texttt{dim=1})\right)$ $\qquad\triangleright$ Rescaled $\mathcal{P}_m \in \mathbb{R}^{N \times N}$

6: $\qquad \mathcal{S}_m = \texttt{Softmax}\left(\sqrt{\frac{h}{d_{\text{model}}}}\mathcal{Q}_m\mathcal{K}_m^{\text{T}}\right)$ $\qquad\qquad\qquad\triangleright \mathcal{S}_m \in \mathbb{R}^{N \times N}$

7: $\qquad \widehat{\mathcal{Z}}_m = \mathcal{S}_m\mathcal{V}_m$ $\qquad\qquad\qquad\qquad\qquad\qquad\triangleright \widehat{\mathcal{Z}}_m \in \mathbb{R}^{N \times \frac{d_{\text{model}}}{h}}$

8: $\widehat{\mathcal{Z}} = \texttt{MLP}_{\texttt{output}}\left(\texttt{Concat}([\widehat{\mathcal{Z}}_1, \cdots, \widehat{\mathcal{Z}}_h], \texttt{dim=1})\right)$ $\qquad\triangleright \widehat{\mathcal{Z}} \in \mathbb{R}^{N \times d_{\text{model}}}$

9: **Return** $\widehat{\mathcal{Z}}$ $\qquad\qquad\qquad\qquad\triangleright$ Keep the $\mathcal{P}_m$ and $\mathcal{S}_m, m = 1, \cdots, h$

---

## C    MORE SHOWCASES

To obtain an intuitive comparison of main results (Table 1), we visualize the criterion of various baselines. Anomaly Transformer can present the **most distinguishable** criterion (Figure 8). Besides,

for the real-world dataset, Anomaly Transformer can also detect the anomalies correctly. Especially for the SWaT dataset (Figure 9(d)), our model can detect the anomalies in the early stage, which is meaningful for real-world applications, such as the early warning of malfunctions.

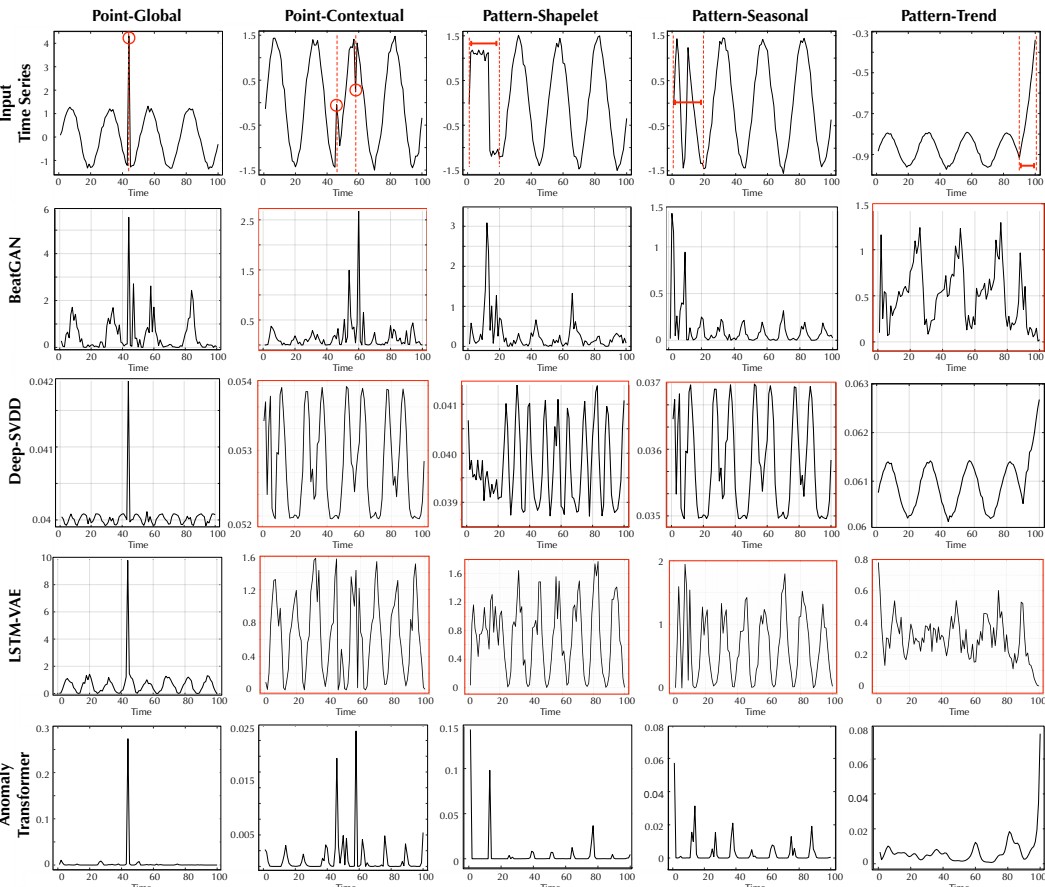

Figure 8: Visualization of learned criterion for the NeurIPS-TS dataset. Anomalies are labeled by red circles and red segments (first row). The failure cases of the baselines are bounded by red boxes.

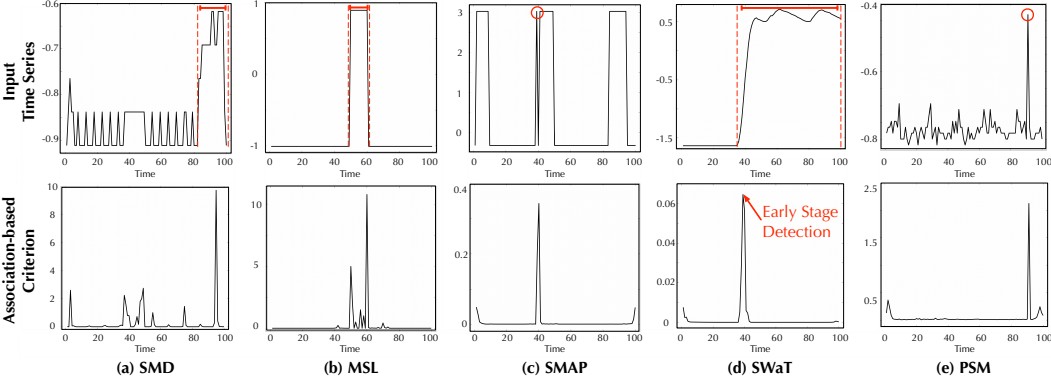

Figure 9: Visualization of the model learned criterion in real-world datasets. We select one dimension of the data for visualization. These showcases are from the test set of corresponding datasets.

## D ABLATION OF ASSOCIATION DISCREPANCY

We present the pseudo code of the calculation in Algorithm 2.

## D.1 ABLATION OF MULTI-LEVEL QUANTIFICATION

We average the association discrepancy from multiple layers for the final results (Equation 6). We further investigate the model performance under the single-layer usage. As shown in Table 4, the multiple-layer design achieves the best, which verifies the effectiveness of multi-level quantification.

Table 4: Model performance under difference selection of model layers for association discrepancy.

| Dataset | SMD | | | MSL | | | SMAP | | | SWaT | | | PSM | | |
|---|---|---|---|---|---|---|---|---|---|---|---|---|---|---|---|
| Metric | P | R | F1 | P | R | F1 | P | R | F1 | P | R | F1 | P | R | F1 |
| layer 1 | 87.15 | 92.87 | 89.92 | 90.36 | 94.11 | 92.19 | 93.65 | 99.03 | 96.26 | 92.61 | 91.92 | 92.27 | 97.20 | 97.50 | 97.35 |
| layer 2 | 87.22 | 95.17 | 91.02 | 90.82 | 92.41 | 91.60 | 93.69 | 98.75 | 96.15 | 92.48 | 92.50 | 92.49 | 96.12 | 98.62 | 97.35 |
| layer 3 | 87.27 | 93.89 | 90.46 | 91.61 | 88.81 | 90.19 | 93.40 | 98.83 | 96.04 | 88.75 | 91.22 | 89.96 | 77.25 | 94.53 | 85.02 |
| Multiple-layer | 89.40 | 95.45 | **92.33** | 92.09 | 95.15 | **93.59** | 94.13 | 99.40 | **96.69** | 91.55 | 96.73 | **94.07** | 96.91 | 98.90 | **97.89** |

## D.2 ABLATION OF STATISTICAL DISTANCE

We select the following widely-used statistical distances to calculate the association discrepancy:

- Symmetrized Kullback–Leibler Divergence (Ours).
- Jensen–Shannon Divergence (JSD).
- Wasserstein Distance (Wasserstein).
- Cross-Entropy (CE).
- L2 Distance (L2).

Table 5: Model performance under different definitions of association discrepancy.

| Dataset | SMD | | | MSL | | | SMAP | | | SWaT | | | PSM | | |
|---|---|---|---|---|---|---|---|---|---|---|---|---|---|---|---|
| Metric | P | R | F1 | P | R | F1 | P | R | F1 | P | R | F1 | P | R | F1 |
| L2 | 85.26 | 74.80 | 79.69 | 85.58 | 81.30 | 83.39 | 91.25 | 56.77 | 70.00 | 79.90 | 87.45 | 83.51 | 70.24 | 96.34 | 81.24 |
| CE | 88.23 | 81.85 | 84.92 | 90.07 | 86.44 | 88.22 | 92.37 | 64.08 | 75.67 | 62.78 | 81.50 | 70.93 | 70.71 | 94.68 | 80.96 |
| Wasserstein | 78.80 | 71.86 | 75.17 | 60.77 | 36.47 | 45.58 | 90.46 | 57.62 | 70.40 | 92.00 | 71.63 | 80.55 | 68.25 | 92.18 | 78.43 |
| JSD | 85.33 | 90.09 | 87.64 | 91.19 | 92.42 | 91.80 | 94.83 | 95.14 | 94.98 | 83.75 | 96.75 | 89.78 | 95.33 | 98.58 | 96.93 |
| Ours | 89.40 | 95.45 | **92.33** | 92.09 | 95.15 | **93.59** | 94.13 | 99.40 | **96.69** | 91.55 | 96.73 | **94.07** | 96.91 | 98.90 | **97.89** |

As shown in Table 5, our proposed definition of association discrepancy still achieves the best performance. We find that both the CE and JSD can provide fairly good results, which are close to our definition in principle and can be used to represent the information gain. The L2 distance is not suitable for the discrepancy, which overlooks the property of discrete distribution. The Wasserstein distance also fails in some datasets. The reason is that the prior-association and series-association are exactly matched in the position indexes. Still, the Wasserstein distance is not calculated point by point and considers the distribution offset, which may bring noises to the optimization and detection.

---

**Algorithm 2** Association Discrepancy $\mathrm{AssDis}(\mathcal{P}, \mathcal{S}; \mathcal{X})$ Calculation (multi-head version).

---

**Input:** time series length $N$; layers number $L$; heads number $h$; prior-association $\mathcal{P}_{\text{all}} \in \mathbb{R}^{L \times h \times N \times N}$; series-association $\mathcal{S}_{\text{all}} \in \mathbb{R}^{L \times h \times N \times N}$;

1: $\mathcal{P}' = \texttt{Mean}(\mathcal{P}, \texttt{dim=1})$            $\triangleright \mathcal{P}' \in \mathbb{R}^{L \times N \times N}$

2: $\mathcal{S}' = \texttt{Mean}(\mathcal{S}, \texttt{dim=1})$            $\triangleright \mathcal{S}' \in \mathbb{R}^{L \times N \times N}$

3: $\mathcal{R}' = \mathrm{KL}\Big((\mathcal{P}', \mathcal{S}'), \texttt{dim=-1}\Big) + \mathrm{KL}\Big((\mathcal{S}', \mathcal{P}'), \texttt{dim=-1}\Big)$     $\triangleright \mathcal{R}' \in \mathbb{R}^{L \times N}$

4: $\mathcal{R} = \texttt{Mean}(\mathcal{R}', \texttt{dim=0})$            $\triangleright \mathcal{R} \in \mathbb{R}^{N \times 1}$

5: **Return** $\mathcal{R}$            $\triangleright$ Represent the association discrepancy of each time point

---

### D.3 ABLATION OF PRIOR-ASSOCIATION

In addition to the Gaussian kernel with a learnable scale parameter, we also try to use the power-law kernel $P(x; \alpha) = x^{-\alpha}$ with a learnable power parameter $\alpha$ for prior-association, which is also a unimodal distribution. As shown in Table 6, power-law kernel can achieve a good performance in most of the datasets. However, because the scale parameter is easier to optimize than the parameter of power, Gaussian kernel still surpasses the power-law kernel consistently.

Table 6: Model performance under different definitions of prior-association. Our Anomaly Transformer adopts the Gaussian kernel as the prior. *Power-law* refers to the power-law kernel.

| Dataset | SMD | | | MSL | | | SMAP | | | SWaT | | | PSM | | |
|---|---|---|---|---|---|---|---|---|---|---|---|---|---|---|---|
| Metric | P | R | F1 | P | R | F1 | P | R | F1 | P | R | F1 | P | R | F1 |
| Power-law | 89.41 | 92.46 | 90.91 | 90.95 | 85.87 | 88.34 | 91.95 | 58.24 | 71.31 | 92.52 | 93.29 | 92.90 | 96.46 | 98.15 | 97.30 |
| Ours | 89.40 | 95.45 | **92.33** | 92.09 | 95.15 | **93.59** | 94.13 | 99.40 | **96.69** | 91.55 | 96.73 | **94.07** | 96.91 | 98.90 | **97.89** |

## E ABLATION OF ASSOCIATION-BASED CRITERION

### E.1 CALCULATION

We present the pseudo-code of association-based criterion in Algorithm 3.

---
**Algorithm 3** Association-based Criterion `AnomalyScore(`$\mathcal{X}$`)` Calculation

---
**Input:** time series length $N$; input time series $\mathcal{X} \in \mathbb{R}^{N \times d}$; reconstruction time series $\widehat{\mathcal{X}} \in \mathbb{R}^{N \times d}$;
   association discrepancy $\text{AssDis}(\mathcal{P}, \mathcal{S}; \mathcal{X}) \in \mathbb{R}^{N \times 1}$;

1: $\mathcal{C}_{\text{AD}} = \texttt{Softmax}(-\text{AssDis}(\mathcal{P}, \mathcal{S}; \mathcal{X}), \texttt{dim=0})$         $\triangleright \mathcal{C}_{\text{AD}} \in \mathbb{R}^{N \times 1}$

2: $\mathcal{C}_{\text{Recon}} = \texttt{Mean}\left((\mathcal{X} - \widehat{\mathcal{X}})^2, \texttt{dim=1}\right)$         $\triangleright \mathcal{C}_{\text{Recon}} \in \mathbb{R}^{N \times 1}$

3: $\mathcal{C} = \mathcal{C}_{\text{AD}} \times \mathcal{C}_{\text{Recon}}$         $\triangleright \mathcal{C} \in \mathbb{R}^{N \times 1}$

4: **Return** $\mathcal{C}$         $\triangleright$ Anomaly score for each time point

---

### E.2 ABLATION OF CRITERION DEFINITION

We explore the model performance under different definitions of anomaly criterion, including the pure association discrepancy, pure reconstruction performance and different combination methods for association discrepancy and reconstruction performance: addition and multiplication.

$$\text{Association Discrepancy: AnomalyScore}(\mathcal{X}) = \text{Softmax}\Big(-\text{AssDis}(\mathcal{P}, \mathcal{S}; \mathcal{X})\Big),$$

$$\text{Reconstruction: AnomalyScore}(\mathcal{X}) = \Big[\|\mathcal{X}_{i,:} - \widehat{\mathcal{X}}_{i,:}\|_2^2\Big]_{i=1,\cdots,N},$$

$$\text{Addition: AnomalyScore}(\mathcal{X}) = \text{Softmax}\Big(-\text{AssDis}(\mathcal{P}, \mathcal{S}; \mathcal{X})\Big) + \Big[\|\mathcal{X}_{i,:} - \widehat{\mathcal{X}}_{i,:}\|_2^2\Big]_{i=1,\cdots,N},$$

$$\text{Multiplication (Ours): AnomalyScore}(\mathcal{X}) = \text{Softmax}\Big(-\text{AssDis}(\mathcal{P}, \mathcal{S}; \mathcal{X})\Big) \odot \Big[\|\mathcal{X}_{i,:} - \widehat{\mathcal{X}}_{i,:}\|_2^2\Big]_{i=1,\cdots,N}.$$
$$(7)$$

From Table 7, we find that directly using our proposed association discrepancy can also achieve a good performance, which surpasses the competitive baseline THOC (Shen et al., 2020) consistently. Besides, the multiplication combination that we used in Equation 6 performs the best, which can bring a better collaboration to the reconstruction performance and association discrepancy.

Table 7: Ablation of criterion definition. We also include the state-of-the-art deep model THOC (Shen et al., 2020) for comparison. *AssDis* and *Recon* represent the pure association discrepancy and the pure reconstruction performance respectively. *Ours* refers to our proposed association-based criterion with the multiplication combination.

| Dataset | SMD | | | MSL | | | SMAP | | | SWaT | | | PSM | | | Avg |
|---|---|---|---|---|---|---|---|---|---|---|---|---|---|---|---|---|
| Metric | P | R | F1 | P | R | F1 | P | R | F1 | P | R | F1 | P | R | F1 | F1(%) |
| THOC | 79.76 | 90.95 | 84.99 | 88.45 | 90.97 | 89.69 | 92.06 | 89.34 | 90.68 | 83.94 | 86.36 | 85.13 | 88.14 | 90.99 | 89.54 | 88.01 |
| Recon | 78.63 | 65.29 | 71.35 | 79.15 | 78.07 | 78.61 | 89.38 | 56.35 | 69.12 | 76.81 | 86.89 | 81.53 | 69.84 | 94.73 | 80.40 | 76.20 |
| AssDis | 86.74 | 88.42 | 87.57 | 91.20 | 89.81 | 90.50 | 91.56 | 90.41 | 90.98 | 97.27 | 89.48 | 93.21 | 97.80 | 93.25 | 95.47 | 91.55 |
| Addition | 77.16 | 70.58 | 73.73 | 88.08 | 87.37 | 87.72 | 91.28 | 55.97 | 69.39 | 84.34 | 81.98 | 83.14 | 97.60 | 97.61 | 97.61 | 82.32 |
| Ours | 89.40 | 95.45 | **92.33** | 92.09 | 95.15 | **93.59** | 94.13 | 99.40 | **96.69** | 91.55 | 96.73 | **94.07** | 96.91 | 98.90 | **97.89** | **94.96** |

## F  CONVERGENCE OF MINIMAX OPTIMIZATION

The total loss of our model (Equation 4) contains two parts: the reconstruction loss and the association discrepancy. Towards a better control of association learning, we adopt a minimax strategy for optimization (Equation 5). During the minimization phase, the optimization trends to minimize the association discrepancy and the reconstruction error. During the maximization phase, the optimization trends to maximize the association discrepancy and minimize the reconstruction error.

We plot the change curve of the above two parts during the training procedure. As shown in Figures 10 and 11, both parts of the total loss can converge within limited iterations on all the five real-world datasets. This nice convergence property is essential for the optimization of our model.

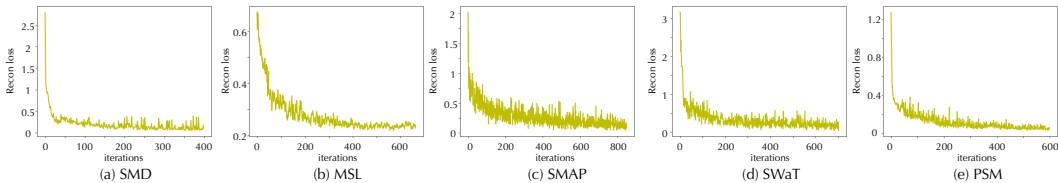

Figure 10: Change curve of reconstruction loss $\|\mathcal{X} - \widehat{\mathcal{X}}\|_F^2$ in real-world datasets during training.

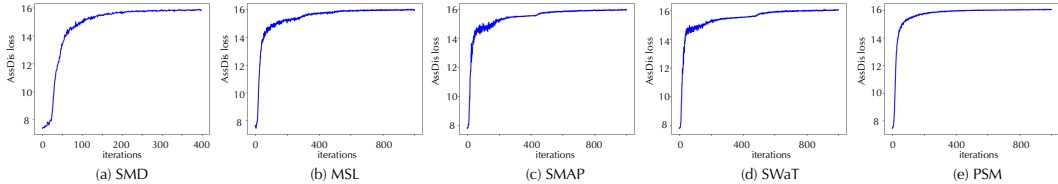

Figure 11: Change curve of association discrepancy $\|\text{AssDis}(\mathcal{P}, \mathcal{S}; \mathcal{X})\|_1$ in real-world datasets during the training process.

## G  MODEL PARAMETER SENSITIVITY

In this paper, we set the hyper-parameters $L$ and $d_{\text{model}}$ following the convention of Transformers (Vaswani et al., 2017; Zhou et al., 2021).

Furthermore, to evaluate model parameter sensitivity, we investigate the performance and efficiency under different choices for the number of layers $L$ and hidden channels $d_{\text{model}}$. Generally, increasing the model size can obtain better results but with larger memory and computation costs.

Table 8: Model performance under different choices of the number of layers $L$.

| Dataset | SMD | | | MSL | | | SMAP | | | SWaT | | | PSM | | |
|---|---|---|---|---|---|---|---|---|---|---|---|---|---|---|---|
| Metric | P | R | F1 | P | R | F1 | P | R | F1 | P | R | F1 | P | R | F1 |
| $L = 1$ | 89.24 | 93.73 | 91.43 | 91.99 | **97.59** | **94.71** | 93.58 | 99.35 | 96.38 | 91.57 | 95.33 | 93.42 | 96.74 | 98.09 | 97.41 |
| $L = 2$ | 89.26 | 94.33 | 91.72 | 91.89 | 94.73 | 93.29 | 93.79 | 98.91 | 96.28 | 92.37 | 94.59 | 93.47 | 97.22 | 98.23 | 97.72 |
| $L = 3$ | 89.40 | 95.45 | 92.33 | **92.09** | 95.15 | 93.59 | **94.13** | **99.40** | **96.69** | 91.55 | **96.73** | **94.07** | 96.91 | **98.90** | **97.89** |
| $L = 4$ | **89.59** | **95.76** | **92.58** | 91.88 | 95.40 | 93.61 | 93.75 | 99.13 | 96.37 | **93.37** | 93.45 | 93.41 | **97.30** | 97.58 | 97.44 |

Table 9: Model performance under different choices of the number of hidden channels $d_{\text{model}}$. *Mem* means the averaged GPU memory cost. *Time* is the averaged running time of 100 iterations during the training process.

| Dataset | SMD | | | MSL | | | SMAP | | | SWaT | | | PSM | | | Mem | Time |
|---|---|---|---|---|---|---|---|---|---|---|---|---|---|---|---|---|---|
| Metric | P | R | F1 | P | R | F1 | P | R | F1 | P | R | F1 | P | R | F1 | (GB) | (s) |
| $d_{\text{model}} = 256$ | 88.83 | 91.82 | 90.30 | 91.96 | **97.60** | **94.70** | 93.74 | 99.47 | 96.52 | **93.91** | 93.99 | 93.95 | **97.38** | 98.16 | 97.77 | 4.9 | 0.12 |
| $d_{\text{model}} = 512$ | 89.40 | 95.45 | 92.33 | **92.09** | 95.15 | 93.59 | **94.13** | 99.40 | **96.69** | 91.55 | **96.73** | **94.07** | 96.91 | **98.90** | **97.89** | 5.5 | 0.15 |
| $d_{\text{model}} = 1024$ | **89.44** | **96.33** | **92.76** | 91.80 | 94.99 | 93.37 | 93.58 | **99.47** | 96.43 | 92.02 | 95.01 | 93.49 | 95.78 | 98.12 | 96.94 | 6.6 | 0.27 |

## H    PROTOCOL OF THRESHOLD SELECTION

Our paper focuses on unsupervised time series anomaly detection. Experimentally, each dataset includes training, validation and testing subsets. Anomalies are only labeled in the testing subset. Thus, we select the hyper-parameters following the **Gap Statistic method (Tibshirani et al., 2001) in K-Means**. Here is the selection procedure:

- After the training phase, we apply the model to the validation subset (without label) and obtain the anomaly scores (Equation 6) of all time points.

- We count the frequency of the anomaly scores in the validation subset. It is observed that the distribution of anomaly scores is separated into two clusters. We find that the cluster with a larger anomaly score contains $r$ time points. And for our model, $r$ is closed to 0.1%, 0.5%, 1% for SWaT, SMD and other datasets respectively (Table 10).

- Due to the size of the test subset being still inaccessible in real-world applications, we have to fix the threshold as a fixed value $\delta$, which can gaurantee that the anomaly scores of $r$ time points in the validation set are larger than $\delta$ and thus detected as anomalies.

Table 10: Statistical results of anomaly score distribution on the validation set. We count the number of time points with corresponding values in several intervals.

(a) SMD, MSL and SWaT datasets.

| Anomaly Score Interval | SMD | MSL | SWaT |
|---|---|---|---|
| $(0, +\infty]$ | 141681 | 11664 | 99000 |
| $[0, 10^{-2}]$ | 140925 | 11537 | 98849 |
| $(10^{-2}, 0.1]$ | 2 | 8 | 17 |
| $(0.1, +\infty]$ | 754 | 119 | 134 |
| Ratio of $(0.1, +\infty]$ | 0.53% | 1.02% | 0.14% |

(b) SMAP and PSM datasets.

| Anomaly Score Interval | SMAP | PSM |
|---|---|---|
| $(0, +\infty]$ | 27037 | 26497 |
| $[0, 10^{-3}]$ | 26732 | 26223 |
| $(10^{-3}, 10^{-2}]$ | 0 | 5 |
| $(10^{-2}, +\infty]$ | 305 | 269 |
| Ratio of $(10^{-2}, +\infty]$ | 1.12% | 1.01% |

Note that, directly setting the $\delta$ is also feasible. According to the intervals in Table 10, we can fix the $\delta$ as 0.1 for the SMD, MSL and SWaT datasets, 0.01 for the SMAP and PSM datasets, which yield a quite close performance to setting $r$.

Table 11: Model performance. *Choose by $\delta$* means that we fix $\delta$ as 0.1 for the SMD, MSL and SWaT datasets, 0.01 for the SMAP and PSM datasets. *Choose by $r$* means that we select $r$ as 0.1% for SWaT, 0.5% for SMD and 1% for the other datasets.

| Dataset | SMD | | | MSL | | | SMAP | | | SWaT | | | PSM | | |
|---|---|---|---|---|---|---|---|---|---|---|---|---|---|---|---|
| Metric | P | R | F1 | P | R | F1 | P | R | F1 | P | R | F1 | P | R | F1 |
| Choose by $\delta$ | 88.65 | 97.17 | **92.71** | 91.86 | 95.15 | 93.47 | 97.69 | 98.24 | **97.96** | 86.02 | 95.01 | 90.29 | 97.69 | 98.24 | **97.96** |
| Choose by $r$ | 89.40 | 95.45 | 92.33 | 92.09 | 95.15 | **93.59** | 94.13 | 99.40 | 96.69 | 91.55 | 96.73 | **94.07** | 96.91 | 98.90 | 97.89 |

In real-world applications, the number of selected anomalies is always decided up to human resources. Under this consideration, setting the number of detected anomalies by the ratio $r$ is more practical and easier to decide according to the available resources.

## I    MORE BASELINES

In addition to the time series anomaly detection methods, the methods for change point detection and time series segmentation can also perform as valuable baselines. Thus, we also include the BOCPD (Adams & MacKay, 2007) and TS-CP2 (Deldari et al., 2021) from change point detection and U-Time (Perslev et al., 2019) from time series segmentation for comparison. Anomaly Transformer still achieves the best performance.

Table 12: Additional quantitative results for Anomaly Transformer (*Ours*) in five real-world datasets. The *P*, *R* and *F1* represent the precision, recall and F1-score (as %) respectively. F1-score is the harmonic mean of precision and recall. For these metrics, a higher value indicates a better performance.

| Dataset | SMD | | | MSL | | | SMAP | | | SWaT | | | PSM | | |
|---|---|---|---|---|---|---|---|---|---|---|---|---|---|---|---|
| Metric | P | R | F1 | P | R | F1 | P | R | F1 | P | R | F1 | P | R | F1 |
| BOCPD | 70.90 | 82.04 | 76.07 | 80.32 | 87.20 | 83.62 | 84.65 | 85.85 | 85.24 | 89.46 | 70.75 | 79.01 | 80.22 | 75.33 | 77.70 |
| TS-CP2 | 87.42 | 66.25 | 75.38 | 86.45 | 68.48 | 76.42 | 87.65 | 83.18 | 85.36 | 81.23 | 74.10 | 77.50 | 82.67 | 78.16 | 80.35 |
| U-Time | 65.95 | 74.75 | 70.07 | 57.20 | 71.66 | 63.62 | 49.71 | 56.18 | 52.75 | 46.20 | 87.94 | 60.58 | 82.85 | 79.34 | 81.06 |
| Ours | 89.40 | 95.45 | **92.33** | 92.09 | 95.15 | **93.59** | 94.13 | 99.40 | **96.69** | 91.55 | 96.73 | **94.07** | 96.91 | 98.90 | **97.89** |

## J    LIMITATIONS AND FUTURE WORK

**Window size**   As shown in the Figure 7 of Appendix A, the model may fail if the window size is too small for association learning. But the Transformers is with quadratic complexity w.r.t. the window size. The trade-off is needed for real-world applications.

**Theoretical analysis**   As a well-established deep model, the performance of Transformers has been explored in previous works. But it is still under-exploring for the theory of complex deep models. In the future, we will explore the theorem of Anomaly Transformer for better justifications in light of classic analysis for autoregression and state space models.

## K    DATASET

Here is the statistical details of experiment datasets.

Table 13: Details of benchmarks. AR represents the truth abnormal proportion of the whole dataset.

| Benchmarks | Applications | Dimension | Window | #Training | #Validation | #Test (labeled) | AR (Truth) |
|---|---|---|---|---|---|---|---|
| SMD | Server | 38 | 100 | 566,724 | 141,681 | 708,420 | 0.042 |
| PSM | Server | 25 | 100 | 105,984 | 26,497 | 87,841 | 0.278 |
| MSL | Space | 55 | 100 | 46,653 | 11,664 | 73,729 | 0.105 |
| SMAP | Space | 25 | 100 | 108,146 | 27,037 | 427,617 | 0.128 |
| SWaT | Water | 51 | 100 | 396,000 | 99,000 | 449,919 | 0.121 |
| NeurIPS-TS | Various Anomalies | 1 | 100 | 20,000 | 10,000 | 20,000 | 0.018 |

## L    UCR DATASET

UCR Dataset is a very challenging and comprehensive dataset provided by the *Multi-dataset Time Series Anomaly Detection Competition* of KDD2021 (Keogh et al., Competition of International Conference on Knowledge Discovery & Data Mining 2021). The whole dataset contains 250 sub-datasets, covering various real-world scenarios. Each sub-dataset of UCR has only one anomaly segment and only has one dimension. These sub-datasets range in length from 6,684 to 900,000 and are pre-divided into training and test sets.

We also experiment on the UCR dataset for a wide evaluation. As show in Table 14, our Anomaly Transformer still achieves the state-of-the-art in this challenging benchmark.

Table 14: Quantitative results in UCR Dataset. *IF* refers to the IsolationForest (2008). *Ours* is our Anomaly Transformer. *P*, *R* and *F1* represent the precison, recall and F1-score (%) respectively.

| Metric | LSTM-VAE | InterFusion | OmniAnomaly | THOC | Deep-SVDD | BeatGAN | LOF | OC-SVM | IF | **Ours** |
|---|---|---|---|---|---|---|---|---|---|---|
| P | 62.08 | 60.74 | 64.21 | 54.61 | 47.08 | 45.20 | 41.47 | 41.14 | 40.77 | 72.80 |
| R | 97.60 | 95.20 | 86.93 | 80.83 | 88.91 | 88.42 | 98.80 | 94.00 | 93.60 | 99.60 |
| F1 | 75.89 | 74.16 | 73.86 | 65.19 | 61.56 | 59.82 | 58.42 | 57.23 | 56.80 | **84.12** |

