# OpenReview forum: "Anomaly Transformer: Time Series Anomaly Detection with Association Discrepancy"
_ICLR.cc/2022/Conference — ICLR 2022 Spotlight_

### Official Review · Reviewer_BKSk · 2021-11-01

**Correctness:** 3
**Technical Novelty And Significance:** 3
**Empirical Novelty And Significance:** 3
**Recommendation:** 8
**Confidence:** 4

**Main Review:**

Update after the discussion with the authors.

The authors have addressed all of my immediate concerns. The paper now looks very strong. I recommend acceptance.

----
The key idea of using Tranformer's self-attention as a measure of anomalousness sounds novel. It is an excellent idea. The proposed architecture featuring a two-branch attention mechanism also looks novel. The novelty seems undisputable, to the best of my knowledge.

The problem is, however, that the paper mostly ignores almost all the existing anomaly detection approaches for **time series**. It obviously does not make sense to use point-wise anomaly detection methods such as SVDD to capture sequential anomalies. Many researchers in this domain may agree that the baseline model in the present context can be the vector autoregressive model, which naturally realizes a particular type of self-attention in the form of the lag-dependent covariance matrices. You can find many works that leverage a dependency graph for anomaly or change detection. I'd also suggest looking at the literature on time-series segmentation.

Another issue is that the paper lacks sound justifications for the proposed approach. Given much theoretical/empirical work with VAR or state-space models (or their neural extensions), we expect a much more understandable derivation of the proposed model.  Many "functions" such as AssDiss lack a proper definition.
 (For example, I didn't understand how the probability distributions had been defined from P^l and S^l --- just writing SoftMax or just showing Eq.(2) does not mean you have defined a distribution for a **matrix**. The Gaussian distribution in Eq.(2) is defined in the entire real domain. But probably, you are on a regular time grid. Apparently, some mathematical inconsistency exists.)

That's not a problem if this were part of API documentation of a software library. But as a technical paper submitted to a top machine learning conference, there may be different expectations.

I think the main idea deserves much more careful and deep thoughts. I encourage the authors to re-do the text to perfect it. I'm sure that the new version will be a great piece of work in the community.


**Summary Of The Paper:**

This paper proposes a new anomaly detection approach based on a Transformer architecture. The main idea is to leverage self-attention to capture the temporal dependency structure of the observations in a sliding window as a measure of anomaly. Since anomalies can be defined as a sequence that is inconsistent with regular ones, the attention matrix is expected to reflect some major aspects of anomalies.

The authors propose a two-branch attention architecture to handle multi-dimensional real-valued time-series data, where prior- and series-attention matrices are computed and utilized in a certain min-max competitive learning framework.

The final anomaly score is defined as the product between the re-construction error and the KL distance between the two attention matrices.

**Summary Of The Review:**

Updated after the discussion
- Novel idea of using the degree of self-attention as a metric of anomalousness.
- New approach of positional encoding based on Gaussian kernels
- Comprehensive empirical comparison with alternative methods

----- Original summary -----
- Novel idea of using the degree of self-attention as a metric of anomalousness.
- Unclear and unjustified descriptions.
- Lack of the critical baseline (major issue in this domain).

---

> ### Author Response · Authors · 2021-11-14
> **Response to Reviewer BKSk [Part 1]**
>
>
> We would like to sincerely thank Reviewer BKSk for providing the detailed review and insightful suggestions. We will further address the points in the revision.
>
> > **Q1:** Critical baselines.
>
> As shown in $\underline{\text{Table 2 of original version}}$, we have provided 10 baseline for extensive comparison. Especially, the LSTM-VAE, BeatGAN, OmniAnomaly, InterFusion and THOC are for time series anomaly detection by **exploring the temporal dependencies with Transformers or RNNs**. All the five baselines are highly related to our paper, where **InterFusion and THOC are previous state-of-the-art methods.**
>
> As per your request, we further expand our set of baselines with **3 new vector autoregressive models**: classic VAR based on ARIMA, LSTM (Hundman et al., 2018) and CL-MPPCA (Tariq et al., 2019) in the $\underline{\text{Related Work, Table 2 of revised paper}}$.
>
> Besides, following your suggestion, we survey the change detection and time-series segmentation areas. Although these two areas are different from time series anomaly detection, we still adopt their widely-used methods to time series anomaly detection as the baselines $\underline{\text{Appendix I of revised paper}}$, including BOCPD (Adams & MacKay, 2007), TS-CP2 (Deldari et al., 2021) and U-Time (Perslev et al., 2019).
>
> With the above effort, in the $\underline{\text{revised paper}}$, **we have provided 18 baselines in total**.
>
> | Year/Types     | Time series anomaly detection (10 baselines)        | General anomaly detection (5 baselines)   | Other time series areas (3 baselines) |
> | -------------- | ------------------------------------------------------- | ---------------------------------------------- | ------------------------------------------ |
> | Recent 3 years | InterFusion, ITAD, THOC, BeatGAN, OmniAnomaly, CL-MPPCA | -                                              | TS-CP2, U-Time                             |
> | Before 2019    | LSTM-VAE, MPPCACD, LSTM,  VAR                           | OC-SVM, so-lationForest, LOF, DAGMM, Deep-SVDD | BOCPD                                      |
>
> Our Anomaly Transformer still achieves the **state-of-the-art performance under the comparison with more baselines**. Due to space limitation, we only provide the F1-score here. The complete results with precision and recall are provided in the $\underline{\text{revised paper}}$.
>
> | F1-score (%)                      | SMD       | MSL       | SMAP      | SWaT      | PSM       |
> | --------------------------------- | --------- | --------- | --------- | --------- | --------- |
> | VAR (vector autoregression)       | 74.08     | 77.90     | 64.83     | 69.34     | 87.13     |
> | LSTM (vector autoregression)      | 81.78     | 83.95     | 83.39     | 84.69     | 82.80     |
> | CL-MPPCA (vector autoregression)  | 79.09     | 80.44     | 72.88     | 79.07     | 71.80     |
> | BOCPD (change detection)          | 76.07     | 83.62     | 85.24     | 79.01     | 77.70     |
> | TS-CP2 (change detection)         | 75.38     | 76.42     | 85.36     | 77.50     | 80.35     |
> | U-Time (time series segmentation) | 70.07     | 63.62     | 52.75     | 60.58     | 81.06     |
> | **Anomaly Transformer**           | **92.33** | **93.59** | **96.69** | **94.07** | **97.89** |
>
>
>
> > **Q2-Part1:** Many "functions" such as $\mathrm{AssDiss}$ lack a proper definition.
>
> Following your suggestion, we have reformulated **all equations** in the $\underline{\text{revised paper}}$. Below is part of the key modifications:
>
> **(1) How to define the probability distributions from $\mathcal{P}^l$ and $\mathcal{S}^l$.**
>
> **Each row of these two matrices is a discrete distribution.** Recall that $\mathcal{P}^l$ and $\mathcal{S}^l$ are the matrices with the shape of $N\times N$. We use each row of $\mathcal{P}^l$ and $\mathcal{S}^l$ to represent the prior-association and the series-association for the corresponding time point respectively. Thus, we do not attempt to define the matrix distribution. We only use the $\mathrm{Softmax(\cdot)}$ and $\mathrm{Scale(\cdot)}$ to normalize the row of $\mathcal{P}^l$ and $\mathcal{S}^l$ to sum to $1$.
>
> - In the $\underline{\text{revised paper}}$, we reorganized the explanation of $\underline{\text{Equation (2)}}$ to clarify that we define the probability distributions from the rows of $\mathcal{P}^l$ and $\mathcal{S}^l$.
>
> - For the definition of $\mathrm{AssDiss}$ ($\underline{\text{Equation (6)}}$), the KL divergence can be calculated between the pair of rows in $\mathcal{P}^l$ and $\mathcal{S}^l$. We rephrased the equation as follows:
>   $$
>   \mathrm{AssDis}(\mathcal{P},\mathcal{S};\mathcal{X})=\bigg[\frac{1}{L}\sum_{l=1}^{L}\Big(\mathrm{KL}(\mathcal{P_{i,:}}^{l}\|\mathcal{S_{i,:}}^{l})+\mathrm{KL}(\mathcal{S_{i,:}}^{l}\|\mathcal{P_{i,:}}^{l})\Big)\bigg]_{i=1,\cdots,N},
>   $$
>   where **$\mathrm{KL}(\cdot\|\cdot)$ is to calculate the KL divergence between the discrete distributions, that is, every row of $\mathcal{P}^l$ and $\mathcal{S}^l$.**

---

> > ### Author Response · Authors · 2021-11-14
> > **Response to Reviewer BKSk [Part 2]**
> >
> >
> > > **Q2-Part2:** Many "functions" such as $\mathrm{AssDiss}$ lack a proper definition.
> >
> > **(2) The prior-association is defined on a regular time grid.**
> >
> > In this paper, we only use the Gaussian distribution as a learnable unimodal prior. In $\underline{\text{Equation (2)}}$, the prior-association is defined by the probability density of Gaussian distribution **at discrete values.**
> > $$
> > \mathcal{P}^{l} = \mathrm{Scale}\Bigg(\left[\frac{1}{\sqrt{2\pi}\sigma_i}\exp\left(-\frac{|j-i|^2}{2\sigma_i^2}\right)\right]_{i,j\in\{1,\cdots,N\}}\Bigg)
> > $$
> > We rephrase the explanation for the above equation as follows: Concretely, for the $i$-th time point, its association weight to the $j$-th point is the probability density of Gaussian distribution $\mathcal{N}(i, \sigma_i^{2})$ at index $|j-i|$. Further, we use $\mathrm{Scale}(\cdot)$ to transform the association weights to the discrete distributions $\mathcal{P}^l$ by dividing the row sum.
> >
> > Also, we have revised all the sentences about the "Gaussian" to make the concept clearer.
> >
> >
> >
> > > **Q3:** Justifications for the proposed approach.
> >
> > (1) Note that both SSM and Transformers are large communities. Our paper is based on the **well-established Transformers** model and the **well-designed self-attention** module.
> >
> > The performance of Transformers has been widely explored or proved in well-known recent papers, such as GPT-3 (best paper award in NeurIPS 2020, about NLP), Informer (best paper award in AAAI 2021, about time series), Swin Transformer (best paper award in ICCV 2021, about computer vision). And **all the above best papers use the empirical studies for justifications, which we have done in even more depth in our paper.**
> >
> > (2) Concretely, as per your request, we have provided the following empirical results for a sufficient justification, which can provide detailed explanations for the design of every module in Anomaly Transformer.
> >
> > - Key Insights (Motivation):
> >   - **Statistics of series-association** are provided in $\underline{\text{Table 4 of original version}}$ to explain the association discrepancy between normal and abnormal time points.
> >   - **Visualization of prior-association** is provided in $\underline{\text{Table 6 of original version}}$ to verify the *adjacent-concentration inductive bias* of anomalies.
> >
> > - Algorithm Design:
> >   - **Different definitions of association discrepancy** are explored in $\underline{\text{Appendix D.1, D.2  of original version}}$ to clarify the design of association discrepancy ($\underline{\text{Equation 3}}$).
> >   - **Ablation of prior distribution selection** is provided in $\underline{\text{Appendix D.3 of revised paper}}$ to explain the reason why we choose the Gaussian distribution as the unimodal prior.
> >   - **Different definitions of detection criterion** are explored in $\underline{\text{Appendix E.2 of original version}}$ to explain the combination of reconstruction error and association discrepancy.
> > - Model Performance:
> >   - **Extensive comparisons** are provided in $\underline{\text{Table 2, Appendix I of revised paper}}$ with 18 baselines, covering various types of methods from both closely and broadly related areas.
> >   - **Complete ablations** are provided in $\underline{\text{Table 3 of original version}}$ to verify the effectiveness of every module in Anomaly Transformer.
> >   - **Showcases** are provided in $\underline{\text{Figures 5, 8, 9}}$ to explain the reason why our proposed criterion is better than reconstruction error.
> >   - **Hyper-parameter sensitivity** is analyzed in $\underline{\text{Appendix A, G}}$.
> >   - **Convergence performance** of minimax optimization is analyzed in $\underline{\text{Appendix F of revised paper}}$.
> >
> > As elaborated above, we have exploited *an exhaustive set of empirical studies* to clarify the motivation, designs of our model, model performance and insight analysis.
> >
> >
> >
> >
> >
> >
> >
> > > **Q4:** Re-do the text of the paper.
> >
> > Following your suggestion, we have re-organized all the equations, provided extensive baselines and conducted more analyses in the $\underline{\text{revised paper}}$. Your valuable questions and suggestions are very helpful for us to polish and improve the paper into a better shape.

---

> > > ### Comment · Reviewer_BKSk · 2021-11-16
> > > **Thank you for your detailed response**
> > >
> > > (1) Regarding "re-do the text", you will need to update the overall context. The use of transformer for detecting association anomalies is novel, but there are many prior works that attempt to detect anomalies as a structural change of multivariate time-series as a dynamic graph. Subsequence-based anomaly/change detection is another thread of popular methods, where a time-series subsequence is treated as a vector or a matrix, and you compute the discrepancy from the reference state. It is a common approach to detecting association or sequential anomalies. You cannot ignore those prior work.
> > >
> > > Note that I am **not** saying that you should empirically compare against those. No. The point is simply that *you need to properly position your work and clarify the scope of the task you are addressing*. Your idea deserves it.
> > >
> > >
> > > (2) Regarding "justifications", again, what I meant is **not** accumulating more empirical evidence from other domains. Many recent successes of the transformer in different domains may have motivated you, but it does not directly justify your method in *your* task. Here is my original comment:
> > > > Given much theoretical/empirical work with VAR or state-space models (or their neural extensions), we expect a much more understandable derivation of the proposed model.
> > >
> > > We have a reasonable understanding of when the VAR model would work and when it would not from a spectral analysis point of view. State-space models are a bit tricky, but we can justify it by pointing out that the mathematical structure of the underlying dynamics of the physical system can be more consistent than VAR models.
> > >
> > > Specifically, for example, you use some sort of the Gram matrix ($\mathcal{QK}^\top$) for the series association. This is something that the subsequence-based approach has used before and imposes a specific way of evaluating the similarity. You may want to explain where your model comes from and why your model is likely to work and when it may fail, in light of existing models whose general properties are known. If you have no clear idea about that, you need to put it as future work.
> > >
> > > (3) Minor comment: Regarding (2), it should be called the Gaussian **kernel**, not the Gaussian distribution. To be a Gaussian distribution, it needs to satisfy the normalization condition of Gaussian, which is integral over $(-\infty,\infty)$.

---

> > > > ### Author Response · Authors · 2021-11-19
> > > > **New Response to Reviewer BKSk [Part 1]**
> > > >
> > > > Many thanks to Reviewer BKSk for the further comments, with detailed explanation and constructive suggestions.
> > > >
> > > > > **Q1:** Update the overall context.
> > > >
> > > > Following your clarified suggestion, we have updated the $\underline{\text{Abstract, Introduction sections in revised paper}}$ with a new review of explicit association modeling methods, including the graph-based methods and subsequence-based methods for anomaly detection.
> > > >
> > > > **In the landscape of time series anomaly detection, we categorize the majority of previous methods into two groups: pointwise representation learning and explicit association modeling.** Here are the key points:
> > > >
> > > > - Pointwise representation learning:
> > > >   - We have reviewed in the $\underline{\text{original version}}$. These methods usually detect anomalies with reconstruction or prediction error. But, it is hard for them to model the intricate temporal patterns.
> > > > - Explicit association modeling (add a new $\underline{\text{Paragraph 3 in the introduction section}}$):
> > > >   - Graph in temporal dimension (Cheng et al., 2008; 2009): Previous methods built the graph with time points as the nodes and detected the anomalies with a random walk. Still, it is hard for these classic methods to learn informative representations and model fine-grained associations since **representation and association are relying on each other**.
> > > >   - Graph in variate dimension (Zhao et al., 2020; Deng & Hooi, 2021): These methods built the graph for each time point among the multiple variables in multivariate time series. They focus on the relationship or causality of multiple variables. **The learned graph is limited to a single time point, which is still insufficient for temporal patterns.** In our paper, we focus on the association on the temporal dimension. Thus, these technical solutions are distinctly different from ours.
> > > >   - Subsequence-based anomaly detection (Boniol & Palpanas, 2020): This setting is to determine **whether the subsequence with a fixed length is abnormal or not**. In contrast, we solve the anomaly detection problem by determining **whether each time point is anomalous or not**. Besides, we learn the fine-grained time-point temporal associations by the Transformers.
> > > >
> > > > Based on your further review, we position our Anomaly Transformer as follows:
> > > >
> > > > | Explicit association modeling | Temporal dimension             | Variate dimension                        |
> > > > | -------------------- | ------------------------ | ------------------------------------ |
> > > > | Graph                | Cheng et al., 2008; 2009 | Zhao et al., 2020; Deng & Hooi, 2021 |
> > > > | Subsequence          | Boniol & Palpanas, 2020  | -                                    |
> > > > | Transformers          | **Ours**                 | -                                    |
> > > >
> > > > - We adopt the Transformers for explicit fine-grained association modeling, renovate it to a two-branch architecture for both prior-association and series-association, and introduce a minimax strategy to make them interplay for mutual improvement.
> > > > - We proposed an association-based criterion based on the observation of association discrepancy, which is co-designed with the Anomaly Transformer.
> > > >
> > > > It is possible that we have missed other works for explicit association modeling. We are open to discuss with more references following the reviewer's additional suggestions.
> > > >
> > > > > Due to the space limitation, we add a **new comment** as [Part 2] for the "Justifications" and "Recorrection Gaussian distribution"

---

> > > > > ### Author Response · Authors · 2021-11-19
> > > > > **New Response to Reviewer BKSk [Part 2]**
> > > > >
> > > > > > **Q2:** Justifications.
> > > > >
> > > > > **(1) Future work**
> > > > >
> > > > > It is an essential and urgent demand to explore the theory for complex deep models.  It is also true that the deep learning community lacks a sound theory for Transformers. Hence we don't have a clear idea yet for the Anomaly Transformer. Our future work includes theoretical study of Anomoly Transformer in light of classic analysis for autoregression and state space models. And we have clarified this as the future work in the $\underline{\text{Conclusion and Future work section in revised paper}}$.
> > > > >
> > > > > **(2) Limitations**
> > > > >
> > > > > Following your guidance, we have provided the limitations of our model that we find experimentally in $\underline{\text{Appendix J in revised paper}}$. Here is the key point:
> > > > >
> > > > > As shown in the $\underline{\text{Figure 7(a) of original version}}$, the model may fail if the window size is too small for association learning.  However, Transformers have a quadratic complexity w.r.t. the window size. Hence a trade-off between window size and computation complexity is needed for real-world applications.
> > > > >
> > > > > **(3) Emperical results.**
> > > > >
> > > > > As listed in the first response, we exploited *an exhaustive set of empirical studies*, including 10 different dimensions with the case study, visualization, statistical results and ablation study. **Although these analyses cannot provide crystal justifications as the theorem, we believe these empirical studies can also make some important aspects of the algorithm clearer.** For example,
> > > > >
> > > > > - Visualization of the showcases in $\underline{\text{Figure 5 of original version}}$ indicates that our association-based design can provide a more distinguishable criterion than reconstruction.
> > > > > - Statical results of series-association in $\underline{\text{Table 3 of original version}}$ show that the minimax can make the normal and abnormal time points more contrastive.
> > > > >
> > > > > Thereby, we add these empirical analyses only to **explain how our model is better than others, or verify if the designed module works as expected or not.**  The future researcher may get some ideas from these empirical studies.
> > > > >
> > > > >
> > > > >
> > > > > > **Q3:** Recorrect the Gaussian distribution to the Gaussian kernel.
> > > > >
> > > > > Thanks for the kind suggestion. We have updated all the texts and figures by replacing the Gaussian distribution to the Gaussian kernel in the $\underline{\text{revised paper}}$.

---

> > > > > > ### Comment · Reviewer_BKSk · 2021-11-19
> > > > > > **Impressive**
> > > > > >
> > > > > > I appreciate your comprehensive reply. It's impressive. I just updated my review. Now I can champion your paper. I may further update the review after reading the revised paper, but basically the discussion has settled.

---

> > > > > > > ### Author Response · Authors · 2021-11-19
> > > > > > > **Thanks for the Response of Reviewer BKSk**
> > > > > > >
> > > > > > > We'd like to thank Reviewer BKsk again for providing an impressive valuable pre-rebuttal review and re-clarifying the review with more detailed explanations.
> > > > > > >
> > > > > > > We'd also thank you for carefully judging our feedback and raising the score! Your constructive suggestions are very helpful for us to improve the paper in a better shape.

---

### Official Review · Reviewer_88ie · 2021-11-01

**Correctness:** 3
**Technical Novelty And Significance:** 3
**Empirical Novelty And Significance:** 3
**Recommendation:** 8
**Confidence:** 4

**Main Review:**

Major Comments:
- Is this approach a generalization / extension of change point detection? That is not mentioned at all.

- The prior association is learned as one that minimized the difference with the series association and the series association is one that is learned to maximize the difference with the prior association? This will promote the series association to learn as non-Gaussian-like of a distribution as possible, while the prior association is finding the closest Gaussian distribution to the series association? Will this minimax strategy not lead to a degeneration where both distributions becomes extremely wide and almost uniform like?

- “The reconstruction loss will guide the series-association to find the most informative associations, such as the adjacent time points of anomalies.” Why are adjacent time points of anomalies most informative? Is this because this “window” of time can be altogether be considered anomalies as opposed to noise from a single out of distribution point?

- The anomaly score just indicated that there is anomalies or not within the N time point window? In order to narrow down where the anomalies are with the time points, do you need to then test smaller windows of time? Wouldn’t smaller windows cause there to be a shortage of data samples needed for learning?

- KL divergence is not very good at measuring differences in the tails of distributions (it does not put enough importance on that area). But shouldn’t the anomalies be in the tails of the distribution as they are rare?

Minor Comments:

Some of the sentences are grammatically strange in the abstract and introduction sections.

**Summary Of The Paper:**

Summary:
This paper is looking at anomalies in time series data. In particular they define anomalies as ones that lie outside some learned Gaussian-like distribution. They propose a minimax objective function that tries to minimize and maximize the discrepancy between a Gaussian distribution with a learned variance parameter and an “empirical” one learned directly through self-attention on the data. They measure discrepancy using the symmetric KL divergence between the two distributions and also use this for their anomaly score.

**Summary Of The Review:**

Overall the paper has goods results and seems sufficiently interesting and novel.

---

> ### Author Response · Authors · 2021-11-14
> **Response to Reviewer 88ie [Part 1]**
>
> We would like to sincerely thank Reviewer 88ie for providing a detailed review with insightful questions.
>
> > **Q1:** Is this approach a generalization/extension of change point detection?
>
> A quick answer is "No". We explain our answer as follows.
>
> (1) Note that both the anomaly detection and the change point detection are mainstream communities of time series. In this paper, we focus on the anomaly detection task, which is widely explored in many papers (see $\underline{\text{Related Work}}$.)
>
> (2) Anomaly detection is different from change point detection because the anomalies can appear in **both forms of segments or single points** (Aminikhanghahi et al., 2016; Wu, 2016).
>
> (3) Change point detection can be a good baseline of anomaly detection. We have provided BOCPD (Adams & MacKay, 2007) and TS-CP2 (Deldari et al., 2021) as baselines, which are classic or advanced algorithms in change point detection. Due to space limitation, we only provide the F1-score here. The complete results with Precision and Recall are deferred to $\underline{\text{Appendix I of revised paper}}$.
>
> | F1-score (%)              | SMD       | MSL       | SMAP      | SWaT      | PSM       |
> | ------------------------- | --------- | --------- | --------- | --------- | --------- |
> | BOCPD (change point detection)  | 76.07     | 83.62     | 85.24     | 79.01     | 77.70     |
> | TS-CP2 (change point detection) | 75.38     | 76.42     | 85.36     | 77.50     | 80.35     |
> | **Anomaly Transformer**   | **92.33** | **93.59** | **96.69** | **94.07** | **97.89** |
>
>
>
> > **Q2:** Will this minimax strategy make the series- and prior-association degenerate to the uniform distribution?
>
> A quick answer is "No". We explain our answer as follows.
>
> (1) As shown in $\underline{\text{Figure 2}}$, the series-association is also constrained by the reconstruction loss. To obtain a better reconstruction, the series-association will not degrease to the non-informative uniform distribution.
>
> (2) We also provide the visualization of learned variance $\sigma$ in prior-association and the statistical result of series-association in $\underline{\text{Figure 6}}$ and $\underline{\text{Table 4}}$ respectively. These results show that neither of these two distributions degenerates the uniform distribution.
>
> (3) We also provide the training curve of association discrepancy in $\underline{\text{Figure 11 of revised paper}}$. During the training process, association discrepancy is quite stable and can converge to a limit value, which can also verify the reliability of our proposed minimax strategy.
>
>
>
> > **Q3:** “The reconstruction loss will guide the series-association to find the most informative associations, such as the adjacent time points of anomalies.” Why are adjacent time points of anomalies more informative?
>
> We use this sentence simply to give an intuitive example. And we have removed this in the $\underline{\text{revised paper}}$.
>
> Our model conducts the point-wise anomaly detection, which means Anomaly Transformer can detect both the segment and point anomalies. Especially for the detection of single point anomalies, we provide the following results:
>
> - For the case study, as shown in $\underline{\text{Figure 5}}$, our model can also detect the single-point anomalies correctly.
>
> - For the model visualization, as shown in $\underline{\text{Figure 6(a)(b)}}$, the learned variance $\sigma$ of the single-point anomalies are relatively small, which makes the association mainly focus on its small neighborhood and can deal with the situation of single-point anomalies.
>
>
>
> > **Q4:** Is the anomaly detection point-wise or window-wise? Some concerns about the window.
>
> Our paper focuses on **point-wise anomaly detection**. As shown in $\underline{\text{Equation (6)}}$, the anomaly score of a length-$N$ time series is in the shape of $N\times 1$,  which can be used to determine whether each time point is abnormal or not.
>
> Window size selection: As stated in the $\underline{\text{implementation details}}$, we split the time series into non-overlapping windows. And we detect the anomalies window by window. We provide the analysis on the window size in $\underline{\text{Appendix A of original version}}$. The window size is selected with the consideration of both the efficiency and temporal information.

---

> > ### Author Response · Authors · 2021-11-14
> > **Response to Reviewer 88ie [Part 2]**
> >
> > > **Q5:** KL divergence is not very good at measuring differences in the tails of distributions.
> >
> > The distributions in our paper refer to the association weights, which describe each time point's temporal dependencies to the other time points. The distribution is not about data frequency, so **there is no tail problem in the series-association or prior-association for anomalies**.
> >
> > As shown in $\underline{\text{Figure 5}}$, based on the association discrepancy, our proposed criterion shows a good distinguishability for normal and abnormal time points.
> >
> > We have provided the comparison with other statistical distances for association discrepancy in $\underline{\text{Appendix D.2 of original version}}$, such as the cross-entropy, Wasserstein distance and Jensen–Shannon Divergence. We find that **KL Divergence shows the best performance compared with other distances.**
> >
> >
> >
> > > **Q6:** Re-check the grammar in the abstract and introduction.
> >
> > Following your suggestion, we have improved the text of these two sections in the $\underline{\text{revised paper}}$.
> >
> >
> >
> > **References**
> >
> > Aminikhanghahi, Samaneh and Diane Joyce Cook. “A survey of methods for time series change point detection.” In *KAIS* 2016
> >
> > Wu, Hu-sheng. “A survey of research on anomaly detection for time series.” In *ICCWAMTIP* 2016

---

> > > ### Author Response · Authors · 2021-11-25
> > > **Requesting feedback from Reviewer 88ie**
> > >
> > > Dear Reviewer 88ie,
> > >
> > > Thanks again for your detailed review and insightful questions.
> > >
> > > We kindly remind that we are entering the final stage of discussion (Nov 22nd-29th). Following your suggestions, we have made every effort to provide all supporting experiments you mentioned and rephrased the paper detail by detail for better clarification.
> > >
> > > If you have any further concerns or questions, please do not hesitate to let us know, and we will respond to them promptly.
> > >
> > > All the best,
> > >
> > >  Authors

---

> > > > ### Comment · Reviewer_88ie · 2021-11-30
> > > > **Updated Final Score**
> > > >
> > > > Thanks for sufficiently addressing my comments. I have adjusted my score. While it does not affect the technical work of the paper and thus I have not penalized it in my score, it is recommended that the paper is edited again for grammar before the final camera-ready version. Some phrase are still very strange such as "Go beyond previous methods, we introduce Transformers"; using "Extending" instead of "Go beyond" would be grammatically correct. Additionally some of the newly added text also suffers from this; "Towards a better reconstruction, anomalies usually decrease the association discrepancy, which will still derive a higher anomaly score." is not a complete sentence.

---

> > > > > ### Author Response · Authors · 2021-11-30
> > > > > **Thanks for the Response of Reviewer 88ie**
> > > > >
> > > > > We would like to thank Reviewer 88ie for providing a detailed valuable pre-rebuttal review. Your detailed suggestions help us a lot in paper revision.
> > > > >
> > > > > Many thanks to Reviewer 88ie for further suggestions in the grammar of the revised paper. Thanks again for your dedication! In the final version, we guarantee to revise the paper and rephrase the sentences.

---

### Official Review · Reviewer_nXE3 · 2021-11-02

**Correctness:** 3
**Technical Novelty And Significance:** 3
**Empirical Novelty And Significance:** 3
**Recommendation:** 6
**Confidence:** 4

**Main Review:**

Strengths:
1. The observation that the abnormal points have a strong local association (or prior association) and a weak global association is interesting.
2. To better model the association discrepancy, the paper proposes a novel min-max association learning strategy to avoid the Gaussian prior reduction problem.
3. Comprehensive evaluation on a variety of anomaly detection datasets demonstrate the effectiveness of the proposed association discrepancy learning strategy.

Weaknesses:
1. What's the convergence property of the min-max strategy?
2. Some details are not very clear.
- For example, in equation (2), what's the shape of $W^l$, and what does the operation $*$ mean?
- Equation (6) is ambigous, $||\mathcal{X}-\hat{\mathcal{X}}||^2_2$ is a scalar, while the AssDis score after Softmax is a N-by-d matrix.
- In the implementation details, for the $r$, why not set $r$ similar to AR of the datasets as shown in Table 1?
- In table 3, for the "Recon" of Anomaly Transformer, if you only use the reconstruction error, then there should be no optimization strategy for the association discrepancy. Why the optimization strategy is "minmax"?
- In figure 5, what are the labels for the y-axis? For the reconstruction and association criteria, are they the values of loss?

**Summary Of The Paper:**

This paper introduces an Anomaly Transformer for detecting anomalies in time series with association discrepancy. This paper introduces two discrepancies: the series association (e.g., period and trend) and the prior association (e.g., the local smoothness or continuity). For abnormal points, these two associations have a small discrepancy and for normal points, there is a large discrepancy between these two associations. This is because abnormal points have a strong local association while the normal points have a global association. In the model, a two-branch strategy is adopted to model the two associations separately. The model is trained by minimizing the reconstruction loss and maximizing the discrepancy. The experimental results demonstrate the effectiveness of the proposed model.

**Summary Of The Review:**

In general, the observation that abnormal points have a large discrepancy between the prior association and the series association is very interesting. A novel Transformer based model is proposed to model this discrepancy. Experiments demonstrate the effectiveness of the proposed method. However, it is unclear about the convergence property for the proposed min-max strategy. Besides, there are many ambiguous details that hinder the readability of the paper.

---

> ### Author Response · Authors · 2021-11-14
> **Response to Reviewer nXE3**
>
>
> Many thanks to Reviewer nXE3 for providing the thorough and insightful comments.
>
> > **Q1:** The convergence property of the minimax strategy.
>
> To address your concern of convergence, we have provided the **training curve of Anomaly Transformer on all the six datasets** in $\underline{\text{Appendix F of revised paper}}$. Experimentally, our algorithm shows a good convergence property.
>
>
>
> > **Q2:** Details of Equation (2).
>
> We have provided more details in the $\underline{\text{revised paper}}$ and expanded the first row of Equation (2) as follows:
>
> $$
>   \text{Initialization:\ } \mathcal{Q},\mathcal{K},\mathcal{V},\sigma=\mathcal{X}^{l-1}W_{\mathcal{Q}}^{l},\mathcal{X}^{l-1}W_{\mathcal{K}}^{l},\mathcal{X}^{l-1}W_{\mathcal{V}}^{l},\mathcal{X}^{l-1}W_{\sigma}^{l}
> $$
>
> where $\mathcal{X}\in\mathbb{R}^{N\times d_{\text{model}}}$, $W_{\mathcal{Q}}^{l},W_{\mathcal{K}}^{l},W_{\mathcal{V}}^{l}\in\mathbb{R}^{d_{\text{model}}\times d_{\text{model}}}, W_{\sigma}^{l}\in\mathbb{R}^{d_{\text{model}\times 1}}$ represent the parameter matrices for $\mathcal{Q},\mathcal{K},\mathcal{V},\sigma$ in the $l$-th layer respectively.
>
>
>
> > **Q3:** Rephrase the Equation (6).
>
> Note that our anomaly score is defined for each time point. And we have updated Equation (6) as follows:
>
> $$
> \mathrm{AnomalyScore}(\mathcal{X})=\mathrm{Softmax}\Big(-\mathrm{AssDis}(\mathcal{P},\mathcal{S};\mathcal{X})\Big)\odot\Big[||\mathcal{X_{i,:}}-\widehat{\mathcal{X_{i,:}}}||^2_{2}\Big]_{i=1,\cdots,N},
> $$
>
> where $\odot$ is the element-wise multiplication. $\mathrm{AnomalyScore}(\mathcal{X})\in\mathbb{R}^{N\times 1}$ represents the anomaly score for each time point. $\mathcal{X}\in\mathbb{R}^{N\times d}$, where $d$ is the dimension of raw series.
>
>
>
> > **Q4:** Why not set $r$ similar to the ground truth anomaly ratio (AR)?
>
> Note that we focus on the **unsupervised** time series anomaly detection setting. Thus, the ground truth anomaly ratio is inaccessible in real-world applications. We provide the ground-truth value of AR in $\underline{\text{Table 1}}$ simply for the introduction of datasets.
>
> To answer your question, we have further provided more details of the selection protocol of $r$ in $\underline{\text{Appendix H of revised paper}}$.
>
>
>
> > **Q5:** "Recon" of Anomaly Transformer in $\underline{\text{Table 3}}$.
>
> As shown in the schema of  $\underline{\text{Table 3}}$, "Recon" only represents the choice of criterion. We fix the other modules of the Anomaly Transformer and only change the selection of criteria for this ablation.
>
> The setting that only uses the reconstruction criterion without optimization and association discrepancy has been provided as the vanilla Transformer in the first row of $\underline{\text{Table 3}}$.
>
>
>
> > **Q6:** The y-axis of $\underline{\text{Figure 5}}$.
>
> As stated in the title and the y-label of $\underline{\text{Figure 5}}$, the y-axis is the value of the criteria. The criterion of Anomaly Transformer is defined in $\underline{\text{Equation (6)}}$.

---

> > ### Author Response · Authors · 2021-11-25
> > **Requesting feedback from Reviewer nXE3**
> >
> > Dear Reviewer nXE3,
> >
> > Thanks again for your dedication in reviewing our paper.
> >
> > We kindly remind that we are entering the final stage of discussion (Nov 22nd-29th). We have made every effort to address your concerns and answer your questions, including providing the training curve for convergence property and rephrasing all the Equations in the $\underline{\text{revised paper}}$.
> >
> > If you have any further concerns or questions, please do not hesitate to let us know, and we will respond to them promptly.
> >
> > All the best,
> >
> > Authors

---

### Official Review · Reviewer_bfbm · 2021-11-02

**Correctness:** 3
**Technical Novelty And Significance:** 3
**Empirical Novelty And Significance:** 3
**Recommendation:** 8
**Confidence:** 5

**Main Review:**

### Strength:
1) The paper is well written and easy to follow
2) Figure 1 and 2 are very intuitive and easy to understand.
3) Detailed empirical analysis.

### Weakness:
1) If I am not wrong then the SMAP and MSL dataset are from [a] but the author cite Su et al. 2019b.
2) There are repetitive ve entries in references. For example Su et al. 2019a and Su et al. 2019b. I suggest that the authors recheck all entries in bibliography carefully.
3) I think it is important to at least provide the reader with different types of methods used for anomaly detection. The authors have done a good job at it but in my humble opinion literature review is still missing some important papers. For example, LSTM based method and SMAP & MSL dataset [a], Dimensionality reduction & clustering method [b], Spatiotemporal method using Convolutional LSTM  [c], Tensor decomposition based method [d].
4) I have several question/concerns/suggestions about the experiment section.

    - How the threshold δ is set?
    - Can you add more details and provide a solid reason on why r=0.5% for SMD, 0.1% for SWAT, 1% for other dataset.
    - The author should discuss the false-positive rate in more details inside the main text as minimizing false-alarm is really important in practical scenarios.
    - What is the reason behind selecting 3 layers for Anomaly Transformer?
    - channel states of hidden state $d_{model}$ is set to 512. Can you provide a reason for selecting this number and also can you discuss the impact of increasing and reducing this number on performance, efficiency, memory etc.
    - In my humble opinion, the term robustness is very loosely used in the paper. I suggest the author to tone down the sentences about robustness.
    -  Building on the previous point, the robustness of Anomaly Transformer is not evaluated against adversarial attacks. Latest research have shown that almost all anomaly detectors such as MSCRED fails against simple FGSM and PGD attack. It would be interesting to see how the robustness of proposed method in those scenarios.
    - Figure 5 and 6, are hard to understand. It suggest that the author add some background shading for anomaly regions and also add the threshold line so that the reader can easily understand how your method is outperforming other methods. Also, Figure 6 need more context and detail in the main text.

  5) I was unable to locate the link to the code repository. It is critical to validate the paper's claims, and one of the simplest ways to do so is to access and run the code. I believe that the authors should consider making the code for their method and empirical experiments available to the reviewers and later release it publicly.


[a] Hundman, Kyle, et al. "Detecting spacecraft anomalies using lstms and nonparametric dynamic thresholding." Proceedings of the 24th ACM SIGKDD international conference on knowledge discovery & data mining. 2018.

[b] Yairi, Takehisa, et al. "A data-driven health monitoring method for satellite housekeeping data based on probabilistic clustering and dimensionality reduction." IEEE Transactions on Aerospace and Electronic Systems 53.3 (2017): 1384-1401.

[c] Tariq, Shahroz, et al. "Detecting anomalies in space using multivariate convolutional LSTM with mixtures of probabilistic PCA." Proceedings of the 25th ACM SIGKDD International Conference on Knowledge Discovery & Data Mining. 2019.

[d] Shin, Youjin, et al. "ITAD: Integrative Tensor-based Anomaly Detection System for Reducing False Positives of Satellite Systems." Proceedings of the 29th ACM International Conference on Information & Knowledge Management. 2020.

**Summary Of The Paper:**

The authors begin by discussing the discrepancy in association between normal and anomalous points and then suggest an anomaly transformer based on an anomaly-attention mechanism that is further improved using a minimax technique. On six different datasets, empirical analysis demonstrates that the proposed method outperforms state-of-the-art anomaly detection methods.

**Summary Of The Review:**

Well written. Good empirical Analysis. But in some places the paper lacks the justifications and proper reasoning. Experiment/results section need some better explanations.

---

> ### Author Response · Authors · 2021-11-14
> **Response to Reviewer bfbm [Part 1]**
>
> We would like to sincerely thank Reviewer bfbm for providing the insightful review and valuable comments.
>
> > **Q1:** Re-check the bibliography.
>
> Following your suggestion, we have corrected the citations of SMAP, MSL datasets, and revised the $\underline{\text{References}}$ section in the $\underline{\text{revised paper}}$.
>
>
>
> > **Q2:** Provide different types of baselines.
>
> As show in $\underline{\text{Table 2 of original version}}$, we have provided 10 baselines for extensive comparison. In the $\underline{\text{revised paper}}$, we have made the evaluation more complete by including 18 baselines in total.
>
> As per your request, we compare the Anomaly Transformer with the following four baselines: LSTM [a], MPPCACD [b], CL-MPPCA [c], ITAD [d]. Note that our paper categorizes the methods by the *anomaly determination criterion*. Thus, we organized the methods in your review into a taxonomy with respect to the model criterion and updated the $\underline{\text{Related Work}}$.
>
> As shown below, **Anomaly Transformer still achieves the state-of-the-art under the comparison with more baselines.** Due to space limitation, we only provide the F1-score here. See $\underline{\text{Table 2 of revised paper}}$ for the complete comparisons, including precision and recall.
>
> | F1-score (%)            | SMD       | MSL       | SMAP      | SWaT      | PSM       |
> | ----------------------- | --------- | --------- | --------- | --------- | --------- |
> | LSTM [a]                | 81.78     | 83.95     | 83.39     | 84.69     | 82.80     |
> | MPPCACD [b]             | 75.02     | 69.95     | 81.73     | 74.73     | 77.29     |
> | CL-MPPCA [c]            | 79.09     | 80.44     | 72.88     | 79.07     | 71.80     |
> | ITAD [d]                | 79.48     | 76.07     | 73.85     | 57.08     | 68.13     |
> | **Anomaly Transformer** | **92.33** | **93.59** | **96.69** | **94.07** | **97.89** |
>
>
>
> > **Q3:** The setting protocol of thresholds $\delta$ and $r$.
>
> Note that our paper focuses on the **unsupervised** time series anomaly detection setting, in which there is no labeled validation set for tuning hyper-parameters. Thus, we select the hyper-parameters following the **Gap Statistic method (Tibshirani et al., 2001) used in K-Means.** We have provided the detailed protocol in $\underline{\text{Appendix H of revised paper}}$. Here are the key points of the protocol:
>
> (1) After the training phase, we apply the model to the validation subset (without label) and obtain the anomaly scores ($\underline{\text{Equation (6)}}$) of all time points.
>
> (2) We count the frequency of the anomaly scores in the validation subset.  It is observed that **the distribution of anomaly scores is separated into two clusters**. This is due to our **minimax association learning** strategy making the anomaly criterion distinguishable. We find that the cluster with a larger anomaly score contains $r$ time points. And for our model, $r$ is close to 0.1%, 0.5%, 1% for SWaT, SMD and other datasets respectively ($\underline{\text{Table 11}}$).
>
> (3) Due to the test subset being still inaccessible in real-world applications, we have to fix the threshold as a fixed value $\delta$, which can guarantee that the anomaly scores of $r$ time points in the validation set are larger than $\delta$ and detected as anomalies.
>
> In summary,
>
> - The $\delta$ is determined based on $r$. We have discussed this in the $\underline{\text{implementation details}}$ and more details are in the $\underline{\text{Appendix H}}$.
> - The $r$ is determined based on the observation of anomaly score frequency.
>
> For sufficient analysis, we also provided the ROC curve to compare the model performance under different choices of $r$ in $\underline{\text{Figure 3 of original version}}$.

---

> > ### Author Response · Authors · 2021-11-14
> > **Response to Reviewer bfbm [Part 2]**
> >
> > > **Q4:** Discuss more the false-positive rate (FPR) in the main text.
> >
> > In the $\underline{\text{original version}}$, we follow the widely-used metrics: precision, recall, and F1-score, which can provide a sound comparison of model performance. As for the **false-positive rate**, we plot the **ROC curve** in $\underline{\text{Figure 3 of original version}}$, which can reflect both the false-positive rate (FPR) and the true-positive rate (TPR) under different settings of $r$.
> >
> > In the $\underline{\text{revised paper}}$, we have updated the main results and model analyses with more discussion about the **false-positive rate**. Here is part of the analyses:
> >
> > - Based on the ROC curve in $\underline{\text{Figure 3}}$, we show that Anomaly Transformer performs well in FPR and TPR.
> > - In the criterion visualization of $\underline{\text{Figure 5}}$, we show that Anomaly Transformer can make the normal time points have a distinctly small anomaly score and further reduce the FPR.
> > - $\underline{\text{Table 4}}$ shows that the minimax optimization can provide a more distinguishable contrast between normal and abnormal time points. Thus, our design in optimization can also reduce the FPR.
> >
> >
> >
> > > **Q5:** Hyper-parameters of Anomaly Transformer.
> >
> > As a well-established model, Transformer has a widely-used hyper-parameter convention (Vaswani et al., 2017) . We follow the previous convention and set the number of layers as 3 and $d_{\text{model}}$ as 512.
> >
> > As per your request, we have provided the ablation study on the numbers of layers and of hidden channels in $\underline{\text{Appendix G of revised paper}}$, **including the analysis of performance, memory and computation efficiency.**
> >
> >
> >
> > > **Q6:** Sentences about robustness.
> >
> > Previous uses of "robustness" is to stress that our model makes the anomalies more distinguishable or performs well on different datasets and for various types of anomalies. Following your suggestion, we have rephrased all the sentences about robustness.
> >
> >
> >
> > > **Q7:** Update the Figures 5 and 6.
> >
> > Following your suggestion, we have updated the $\underline{\text{Figure 5}}$ with background shading and threshold line.
> >
> > We also have re-done the analysis of  $\underline{\text{Figure 6}}$ with more details in the $\underline{\text{revised paper}}$.
> >
> >
> >
> > > **Q8:** Code release.
> >
> > We have provided the pseudo-code of our algorithm in $\underline{\text{Appendix B, D of original version}}$. Every detail about tensor shape and model layers is included in the pseudo-code. **As per our tradition, we guarantee to make the code public upon paper acceptance.**
> >
> >
> >
> > **References**
> >
> > Robert Tibshirani, Guenther Walther, and Trevor Hastie.  Estimating the number of clusters in a dataset via thegap statistic. J. R. Stat. Soc. (Series B), 2001.
> >
> > Ashish Vaswani, Noam Shazeer, Niki Parmar, Jakob Uszkoreit, Llion Jones, Aidan N Gomez, Łukasz Kaiser,and Illia Polosukhin. Attention is all you need. In NeurIPS, 2017.

---

> > > ### Comment · Reviewer_bfbm · 2021-11-22
> > > **Thank you for detailed response**
> > >
> > > The authors address each of my concerns in detail.
> > >
> > > Additionally, I reviewed the revised draft. It appears to be in much better shape, and the scope has been significantly clarified.
> > >
> > > I'll revise my rating.

---

> > > > ### Author Response · Authors · 2021-11-22
> > > > **Thanks for the Response of Reviewer bfbm**
> > > >
> > > > Many thanks to Reviewer bfbm for providing an impressively insightful pre-rebuttal review, which has enabled us to make an effective response. Your detailed suggestions help us a lot. And we guarantee to make the code public upon paper acceptance.

---

### Author Response · Authors · 2021-11-15
**Summary of Revisions**

We sincerely thank all the reviewers for their insightful reviews and valuable comments, which are instructive for us to improve our paper further.



This paper detects time series anomalies based on the Association Discrepancy that is inherently normal-abnormal distinguishable. From this new association-based dimension, we propose the Anomaly Transformer with a two-branch Anomaly-Attention mechanism to compute the Association Discrepancy. A minimax strategy is devised to amplify the normal-abnormal distinguishability of the Association Discrepancy and further derive a new association-based criterion. Anomaly Transformer achieves the state-of-the-art on six benchmarks for three real applications.



The reviewers generally held positive opinions of our paper, in that using Transformer's self-attention as a measure for detection is “**an excellent idea**”, this paper is “**sufficiently interesting and novel**” and “**well written**”, our model “**outperforms state-of-the-art** **anomaly detection methods**”, the experiment provides “**detailed and good empirical analysis**”.



The reviewers also raised insightful and constructive concerns. We made every effort to address all the concerns by providing sufficient evidence and requested results. See $\underline{\text{revised paper}}$ and $\underline{\text{Appendix}}$ for details. Here is the summary of the revisions:



- **Formalization of equations** **(Reviewers nXE3, 88ie, BKSk):** For clearness, we updated all the equations in the revised paper by re-formalizing them to a point-wise expression, as well as their corresponding explanation. Based on this, we further emphasized that the prior-association, series-association and anomaly score are point-wise.
- **Baselines (Reviewers bfbm, BKSk):** We further enlarged our comparison from 10 baselines to 18 baselines in the revised paper, covering the advanced deep models and classic detection methods, even including 3 baselines from change detection and series segmentation. Under the extensive comparison, Anomaly Transformer still provides the best performance.
- **Model justifications (Reviewer BKSk):** We exploited *an exhaustive set of empirical studies*, including 10 different dimensions with the case study, visualization, statistical results and ablation study. The revised paper covers every detail of the motivation, designs of our model, model performance and insight analysis.
- **Experiment results explanation (Reviewers bfbm):** We rephrased the experiment section for more discussion about the false-positive rate and updated the visualization figures for better intuition.
- **Model parameter ablation (Reviewers bfbm):** We added a comprehensive ablation for the number of model layers and hidden channels.
- **Threshold selection (Reviewers bfbm, nXE3):** We further clarify that we followed the Gap Statistic method in K-Means for threshold selection. Also, we elaborated every detailed step in the revised paper.
- **Convergence of minimax optimization (Reviewers nXE3, 88ie):** We provided the training curve for all five real-world datasets. The training curve proved that our method could show a good convergence performance.



The valuable suggestions from reviewers are very helpful for us to revise the paper to a better shape. We'd be very happy to answer any further questions.

---

### Decision · Program_Chairs · 2022-01-20

**Decision:**

Accept (Spotlight)

**Comment:**

The paper proposed a novel approach that leverages the discrepancies between the (global) series association and the (local) prior association for detecting anomalies in time series. The authors provided detailed empirical support to motivate the above detection criterion, and introduced a two-branch attention architecture for modeling the discrepancies and establishing an anomaly score.

All reviewers acknowledge the technical novelty of this work (including the key insight of modeling anomalousness with Transformer’s self-attention and concrete training mechanism via a minimax optimization process) as well as the comprehensiveness of the empirical study.

Meanwhile, there were some concerns in the positioning of the work, in particular in the clarity in connection to related work, and some reviews concern the clarity of the presentation (e.g. missing some details in experimental results), and the clarity of the exposition of the training process. The authors provided effective feedback during the discussion phase, which helped clarify many of the above concerns. All reviewers agree that the revision makes a solid paper and unanimously recommend acceptance of this work.

The authors are strongly encouraged to take into account the feedback from the discussion phase to further improve the clarity concerning the technical details as well as the reproducibility of the results.